# Optimistic Task Inference for Behavior Foundation Models

**Thomas Rupf**[1]    **Marco Bagatella**[12]    **Marin Vlastelica**[1]    **Andreas Krause**[1]

[1]ETH Zürich, Switzerland    [2]Max Planck Institute for Intelligent Systems, Germany

`{thrupf,mbagatella,mvlastelica,krausea}@ethz.ch`

## Abstract

Behavior Foundation Models (BFMs) are capable of retrieving high-performing policy for any reward function specified directly at test-time, commonly referred to as zero-shot reinforcement learning (RL). While this is a very efficient process in terms of compute, it can be less so in terms of data: as a standard assumption, BFMs require computing rewards over a non-negligible inference dataset, assuming either access to a functional form of rewards, or significant labeling efforts. To alleviate these limitations, we tackle the problem of task inference purely through interaction with the environment at test-time. We propose OpTI-BFM, an optimistic decision criterion that directly models uncertainty over reward functions and guides BFMs in data collection for task inference. Formally, we provide a regret bound for well-trained BFMs through a direct connection to upper-confidence algorithms for linear bandits. Empirically, we evaluate OpTI-BFM on established zero-shot benchmarks, and observe that it enables successor-features-based BFMs to identify and optimize an unseen reward function in a handful of episodes with minimal compute overhead.[1]

## 1 Introduction

Zero-shot reinforcement learning (Touati et al., 2023) has gradually gained relevance as a powerful generalization of standard, single-reward RL (Sutton & Barto, 2018; Silver et al., 2021). Zero-shot agents are designed to distill optimal policies for a set of reward functions from unlabeled, offline data (Touati & Ollivier, 2021; Agarwal et al., 2024; Park et al., 2024b; Jajoo et al., 2025). As these methods scale to more complex and broader environments, they are often referred to as Behavior Foundation Models (BFMs) (Park et al., 2024b; Tirinzoni et al., 2025). At their core, the majority of BFMs rely on Universal Successor Features (Ma et al. (2020), USFs). These methods build upon explicit representations of states (*features*): given a set of policies, the expected discounted sum of features along each policy's trajectory may be estimated completely offline (e.g., through TD learning) (Dayan, 1993; Barreto et al., 2017). As long as a reward function lies within the span of features, zero-shot policy evaluation can be performed through a simple scalar product, which in turn enables zero-shot policy improvement, i.e., each policy is implicitly paired with a reward function and updated towards optimality (Touati & Ollivier, 2021).

Once this unsupervised pre-training phase is complete, USFs yield a set of learned policies. Given a reward function that is linear in the features, its optimal policy is by construction indexed by the linear weights describing the reward function in the basis of features, which can in turn be interpreted as *task embeddings*. The process of finding the optimal policy, which we refer to as *task inference*, thus corresponds to solving a linear regression problem. While this process is remarkably efficient in terms of compute, it retains strict requirements in terms of data: a dataset of labeled (state, reward) pairs needs to be provided. In simple settings, this necessity is not particularly problematic: BFMs have been largely trained in simulations (Touati et al., 2023; Park et al., 2024b; Tirinzoni et al., 2025), which makes generating reward labels for the pre-training dataset, or additional data, particularly convenient. Realistically, however, (i) the pre-training dataset might be unavailable or proprietary and, most importantly, (ii) labeling states with rewards might incur significant costs. For instance,

---

[1]Code is available at `https://github.com/ThomasRupf/opti-bfm`.

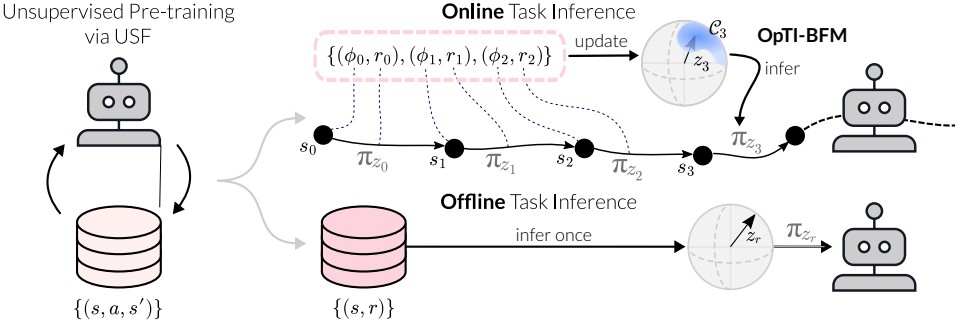

Figure 1: In contrast to the standard offline task inference pipeline for BFMs, we explore an alternative online framework: instead of producing a point-estimate of the task embedding from an existing dataset, we actively collect data to build a belief over task embeddings, which results in milder labeling requirements and fast retrieval of a near-optimal policy.

when BFMs are pre-trained directly from pixels, evaluating success from a single image is not a straightforward or cheap operation.

In order to alleviate these issues, we explore an alternative framework for task inference, which instead aims to collect a small amount of data directly during deployment (see Figure 1). This removes the need to access the pre-training dataset (i) and may require fewer labels (ii), as the data can be collected *actively*. To navigate this setting, we propose OpTI-BFM, a decision criterion that curates a sequence of task embeddings. Starting from an uninformed prior, OpTI-BFM leverages the linear relationship between features and rewards to update its belief over the space of rewards. A BFM conditioned on meaningfully chosen task embeddings can then interact with the environment for a few steps: labeling the states that are visited allows accurate task estimation with minimal labeling efforts. Crucially, if the underlying BFM is well-trained, we show that control reduces to a bandit problem over tasks. Under this assumption, we can provide regret guarantees for OpTI-BFM. When considering established zero-shot benchmarks in the Deepmind Control Suite (DMC) (Tassa et al., 2018), we find that OpTI-BFM requires only a handful of episodes to correctly identify the task, eventually matching or surpassing the performance that the standard offline reward inference pipeline achieves with significantly more data.

## 2 PRELIMINARIES

**Notation** We model the environment as a reward-free Markov Decision Process (MDP) $\mathcal{M} = (S, A, P, \mu, \gamma)$ where $S$ and $A$ are state and action spaces, respectively, $P(ds'|s, a)$ is a probability measure describing the likelihood of transitions, $\mu(ds)$ is a measure describing the initial state distribution, and $\gamma$ is a discount factor. Given a policy $\pi : S \to \Delta(A)$ and a state-action pair $(s_0, a_0) \in S \times A$ we use $\mathbb{E}[\cdot \mid s_0, a_0, \pi]$ to denote expectations w.r.t. state-action sequences $((s_t, a_t))_{t \geq 0}$ defined by sampling $a_t \sim \pi(s_t)$ and $s_{t+1} \sim P(s_t, a_t)$. For a given reward function $r : S \to \mathbb{R}$, we define the policy's state-action value function as the discounted sum of future rewards $Q_r^\pi(s_0, a_0) = \sum_{t \geq 0} \gamma^t \mathbb{E}[r(s_t) \mid s_0, a_0, \pi]$.

**Successor Features and Behavior Foundation Models** For a specific feature map $\phi : S \to \mathbb{R}^d$, Successor Features (Barreto et al. (2017), SFs) generalize state-action value functions by modeling the expected discounted sum of *features* under a policy $\pi$:

$$\psi^\pi(s_0, a_0) = \sum_{t \geq 0} \gamma^t \mathbb{E}[\phi(s_t) \mid s_0, a_0, \pi]. \tag{1}$$

SFs allow zero-shot policy evaluation for any reward that lies in the span of the features: if $r(s) = z^\top \phi(s)$ for some $z \in \mathbb{R}^d$, the Q-function is a linear function of SFs:

$$Q_r^\pi(s_0, a_0) = \sum_{t \geq 0} \gamma^t \mathbb{E}[r(s_t) \mid s_0, a_0, \pi] = \sum_{t \geq 0} \gamma^t \mathbb{E}[z^\top \phi(s_t) \mid s_0, a_0, \pi] = z^\top \psi^\pi(s_0, a_0) \tag{2}$$

A similar structure also holds for value functions: $V_r^\pi(s) = z^\top \psi^\pi(s)$, when defining $\psi^\pi(s) = \mathbb{E}_{a\sim\pi(\cdot|s)}\psi^\pi(s,a)$. Behavior Foundation Models (BFMs) generally[2] capitalize on the opportunity of evaluating policies for multiple reward functions, by additionally learning a family of parameterized policies $(\pi_z)_{z\in\mathcal{Z}}$ with respect to all rewards in the span of features $\phi$ (Borsa et al., 2018; Touati & Ollivier, 2021; Park et al., 2024b; Agarwal et al., 2025b). Concretely, BFMs train a family of parameterized policies so that each policy $\pi_z$ is optimal for the reward function $r(s) = \phi(s)^\top z$:

$$\pi_z(a|s) \in \arg\max_a \psi^{\pi_z}(s,a)^\top z \quad \text{for each } z \in \mathcal{Z}. \tag{3}$$

In practice, the set of policies, their SFs, and features $\phi$ may be represented through function approximation: $\pi_z(a|s) \approx \pi_\xi(a|s,z)$, $\psi^{\pi_z}(s,a) \approx \psi(s,a,z)$, and $\phi(s) \approx \phi(s)$. $\mathcal{Z}$ is a *low-dimensional* space, whose elements can be seen as *task embeddings*, as they represent reward functions in the feature basis. The low dimensionality of task embeddings is a key component enabling efficient task inference in this work.

Given a reward function $r$ at inference, the task embedding $z_r$ parameterizing the optimal learned policy $\pi_{z_r}$ is found by minimizing the residual between $r(s)$ and $\phi(s)^\top z$ (i.e., *projecting* $r$ onto the span of $\phi$). Given a task inference dataset $\mathcal{D} = (s_i)_{i=1}^N$ this may be solved in closed form:

$$z_r = \arg\min_z \mathbb{E}_{s\sim\mathcal{D}}[(r(s) - z^\top \phi(s))^2] = \text{Cov}_\mathcal{D}(\phi)^{-1} \mathbb{E}_{s\sim\mathcal{D}}[\phi(s)r(s)]. \tag{4}$$

where $\text{Cov}_\mathcal{D}(\phi)^{-1} = \mathbb{E}_{s,s'\sim\mathcal{D}}[\phi(s)\phi(s')^\top]$. This process of mapping from a reward function $r$ to an (approximately) optimal policy $\pi_{z_r}$ is what makes BFM based on USFs capable of zero-shot RL (Touati et al., 2023), i.e., they can produce an optimal policy for a previously unseen reward function. When the expectation over $\mathcal{D}$ is computed exactly, and $r$ lies in the span of $\phi$, then $\pi_{z_r}$ is guaranteed to be the optimal policy (Touati & Ollivier, 2021). However, in practice, the expectation is approximated through sampling, which requires (i) the availability of the task inference dataset $\mathcal{D}$ (potentially a subset of the pre-training data) and (ii) providing reward labels for each state $s \in \mathcal{D}$. As this can be an expensive operation, potentially requiring human supervision, we will propose an alternative online framework for retrieving $z_r$.

## 3 OPTIMISTIC TASK INFERENCE FOR BEHAVIOR FOUNDATION MODELS

### 3.1 SETTING: TASK INFERENCE AT TEST-TIME

We consider an alternative framework for task inference in BFMs, designed to remove the necessity for storing pre-training data and, principally, to decrease the required number of reward labels. We focus on an online setting, in which the agent can update the task embedding $z$ during deployment, and directly control the collection of the data used to estimate $z$. While the choice of $z$ will be uninformed at the beginning, it will ideally be possible to rapidly identify the correct task, and thus select the $z$ that retrieves the optimal policy, i.e., that coincides with the true task embedding.

More formally, we start from a pre-trained USF-based BFM, providing a set of parameterized policies $(\pi_z)_{z\in\mathcal{Z}}$, as well as SF estimates $\psi^{\pi_z}$ of features $\phi$. While we will now consider finite-horizon SF estimates $\psi^{\pi_z}(s_0) = \sum_{t=0}^{H-1}\gamma^t \mathbb{E}[\phi(s_t) \mid s_0, \pi_z]$ to streamline the presentation and analysis, we remark that the algorithm can be easily instantiated in infinite-horizon settings, as is done in our empirical evaluation. The agent interacts with the environment in an episodic setting with horizon $H$ and initial state distribution $\mu_0$; instead of directly selecting actions, it will select a task embedding $z_t$ at each step $t$, and execute an action sampled from the respective policy $a_t \sim \pi_{z_t}(\cdot|s_t)$. This action will result in observing a new state $s_{t+1}$, as well as the reward $r_t$ of this transition, which constitutes the only source of information about the task[3].

We define the discounted return of the $k$-th episode as

$$\hat{G}_k = \sum_{t=0}^{H-1}\gamma^t r(s_{kH+t}). \tag{5}$$

---

[2]There are BFMs that are not based on USFs. We refer the reader to Agarwal et al. (2025a) for a comprehensive overview.

[3]We assume that each environment interaction provides a reward label; if the agent can additionally control when to request a reward label, more efficient schemes are possible, see Appendix B.2

where $s_{kH} \sim \mu_0$, $a_t \sim \pi_{z_t}(\cdot|s_t)$ and $s_{t+1} \sim P(\cdot|s_t, a_t)$. The goal of the agent is now simply to minimize the expected regret over $n$ episodes

$$R_n = \mathbb{E}\left[\sum_{k=0}^{n-1} \hat{G}_k^{\star} - \hat{G}_k\right] \tag{6}$$

where $\hat{G}_k^{\star}$ denotes the discounted return choosing $z_r$ in each step, and the expectation is w.r.t. $\mu_0$, the MDP dynamics, the action distributions, and the choices of the task embeddings. Intuitively, the agent needs to follow a decision rule that maps the observed history of states and rewards to a task embedding: $(s_0, r_0, \ldots s_t, r_t) \to z_{t+1} \in \mathcal{Z}$. The selected task embeddings should induce informative trajectories with respect to the reward function, while largely avoiding suboptimal behavior. This setting is reminiscent of well-developed research directions, namely exploration in the space of behavioral priors (Singh et al., 2020) and fast adaptation (Sikchi et al., 2025); however, existing methods ignore the underlying structure connecting rewards and task embeddings, which we will show may be leveraged to efficiently achieve near-optimal performance.

## 3.2 METHOD: OPTI-BFM

Our method leverages a core feature of BFMs: for well-trained USFs, the expectation of the return of the $k$-th episode (Eq. (5)) from an initial state $s_0$ is approximately linear w.r.t. the successor features of the policy conditioned on the $k$-th task embedding $z_k$: $\psi(s_0, z_k)^{\top} z_r \approx \mathbb{E}[\hat{G}_k|s_0, \pi_{z_k}]$, where $z_r$ is the optimal task embedding, and initially unknown. This simple property has significant implication: Policy search reduces to online optimization of a linear function, which has been extensively studied in the bandit literature (Dani et al., 2008; Abbasi-Yadkori et al., 2011). Building upon these fundamental results, we propose **Op**timistic **T**ask **I**nference for **BFM**s (OpTI-BFM) in order to efficiently explore the space of behaviors while controlling suboptimality.

Interestingly, the same approximately linear relationship that connects SFs and returns, also exists between features and rewards: $\phi(s)^{\top} z_r \approx r(s)$. Following the latter property, OpTI-BFM keeps track of a least-squares estimate of $z_r$ given the previously observed transitions

$$\hat{z}_t = \arg\min_{z \in \mathcal{Z}} \sum_{i=0}^{t} \left(r_i - \phi(s_i)^{\top} z\right)^2 + \lambda \|z\|_2^2 = \left(\lambda I_d + \sum_{i=0}^{t} \phi(s_i)\phi(s_i)^{\top}\right)^{-1} \sum_{i=0}^{t} \phi(s_i)\, r_i \tag{7}$$

using $l_2$-regularization to ensure the inverse exists. Rewriting this as

$$\hat{z}_t = V_t^{-1}\sum_{i=0}^{t}\phi(s_i)\, r_i, \quad \text{where} \quad V_t = \lambda I_d + \sum_{i=0}^{t}\phi(s_i)\phi(s_i)^{\top} \tag{8}$$

allows OpTI-BFM to not only track the mean estimate $\hat{z}_t$, but also a confidence ellipsoid $\mathcal{C}_t$ around $\hat{z}_t$ that contains the true task embedding $z_r$ with high probability:

$$\mathcal{C}_t = \left\{z \in \mathbb{R}^d : \left\|z - \hat{z}_{t-1}\right\|_{V_{t-1}} \le \beta_t\right\}, \tag{9}$$

where $\beta_t$ controls the Mahalanobis distance. The estimation of confidence sets allows *optimistic* behavior in each step by choosing the task embedding $z_t$ which is believed to attain the largest return among those in the confidence set:

$$z_t \in \arg\max_{z \in \mathcal{Z}} \max_{w \in \mathcal{C}_t} w^{\top}\psi(s_t, z). \tag{10}$$

Intuitively, this procedure conditions the BFM on the most "promising" task embedding among those that are compatible with rewards observed so far. Note that this algorithm has one crucial difference to Upper Confidence Bound (UCB)-based algorithms for linear contextual bandits (Abbasi-Yadkori et al., 2011; Dani et al., 2008), as two different contexts are involved: the features $\phi$, which are used for online regression and for estimating the confidence interval, and the successor features $\psi$, which are instead used in the acquisition function in Eq. (10). We will discuss that using $\phi$ for regression results in tighter estimates in Appendix A.2.

Algorithm 1 instantiates OpTI-BFM for online task inference. We will establish guarantees for OpTI-BFM in the next section, and then describe how Eq. 10 may be optimized in practice, or avoided altogether with a Thompson Sampling (TS) variant, among others.

---

**Algorithm 1** One episode of online task inference with OpTI-BFM

---

**Require:** BFM with $\psi^{\pi_z}(s, a)$, $\phi(s)$, and $\pi_z(a|s)$, starting state $s_0 \sim \mu_0$, online Least Squares estimator $(\hat{z}_{n-1}, V_{n-1})$ (potentially initialized with past experience)
  **for** $t = 0, \ldots, H - 1$ **do**
    Find $z_t \in \arg \max_{z \in \mathcal{Z}} \max_{w \in \mathcal{C}_{n+t}} w^\top \psi(s_t, z)$          ▷ Optimism w.r.t. cumulative reward.
    Execute action $a_t \sim \pi_{z_t}(\cdot|s_t)$
    Observe reward $r_t$, next state $s_{t+1}$
    Update $(\hat{z}_{n+t-1}, V_{n+t-1})$ with $(\phi(s_t), r_t)$ through Eq. 8 ▷ Update based on reward-feedback.

---

### 3.3 GUARANTEES

Leveraging a direct connection to principled algorithms for linear bandits (Dani et al., 2008; Abbasi-Yadkori et al., 2011), we provide regret guarantees for OpTI-BFM. We note that this is a crucial property for online task inference, which could otherwise fail to gather informative data, and converge to suboptimal solutions. For simplicity, we study a variant of OpTI-BFM that only updates its decision rule at the beginning of each episode, i.e., we have $z_t = z_{t-1}$ for $t \notin \{kH\}_{k=0}^{\infty}$[4]. This additional constraint allows us to leverage results from linear contextual bandit literature.

We can show that OpTI-BFM approaches the performance of the optimal policy $\pi_{z_r}$ (Eq. (4)) under the following assumptions:

(A1) Perfect USF for our setting: for every $(s, a, z) \in \mathcal{S} \times \mathcal{A} \times \mathcal{Z}$ we have $\psi(s_0, a_0, z) = \sum_{t=0}^{H-1} \gamma^t \mathbb{E}[\phi(s_t) \mid \pi_z, s_0, a_0]$ and $\pi_z(a|s) > 0 \implies a \in \arg \max_{a \in \mathcal{A}} \psi(s, a, z)^\top z$.

(A2) Linear Reward: $r$ is in the span of features $\phi$ up to i.i.d. mean-zero $\sigma$-subgaussian noise $\eta_t$, i.e. $r(s_t) = \phi(s_t)^\top z_r + \eta_t$

(A3) Optimization Oracle: the OpTI-BFM objective, Eq. (10), can be computed exactly.

(A4) Bounded norms: $\|z_r\|_2 \le S$ and $\|\phi(\cdot)\|_2 \le L$ for some $S > 0$ and $L > 0$.

Assumptions (A1) and (A2) are instrumental to recovering theoretical guarantees, but we found OpTI-BFM to perform well even when they are violated (see Section 5). Note that for a sufficiently large horizon $H$ the mismatch between the finite discounted sum of features we assume here and the infinite one we have in practice is negligible: the $l_2$-error is by $L\gamma^H/(1 - \gamma)$. Assumption (A3) may be empirically motivated by the efficiency of finding a good approximate solution (see Section 4). In this setting, we can show that OpTI-BFM has sublinear regret.

**Proposition 1.** *(informal) Under assumptions A1-A4, in an episodic discounted MDP, if OpTI-BFM (Algorithm 1) only updates $z_t$ at the start of each episode, it incurs an expected regret of $R_n \le \tilde{\mathcal{O}}(d\sqrt{n})$.*

*Proof.* We prove a formal version, Proposition 5, in Appendix A. The proof is similar to the standard regret bounds for LinUCB/OFUL (Dani et al., 2008; Abbasi-Yadkori et al., 2011) except that the confidence interval is updated $H$-times per step with features that differ from the context features of the bandit. ☐

## 4 PRACTICAL ALGORITHM

Having introduced and analyzed OpTI-BFM, we now discuss a practical implementation, and present some additional variants [5].

The main challenge for a practical implementation of OpTI-BFM lies in optimizing the decision criterion in Eq. (10), involving two continuous spaces $\mathcal{Z}$ and $\mathcal{C}_t$, and a highly non-linear map $z \mapsto \psi(\cdot, z)$. Fortunately, BFMs are pre-trained such that that $w \approx \arg \max_{z \in \mathcal{Z}} w^\top \psi(s_t, z)$, i.e. the optimal policy for a task described by $w$ is the one conditioned on $w$ itself. In practice, as training is not perfect, we do not strictly rely on this property, and still search $z$ over $\mathcal{C}_t$ instead of $\mathcal{Z}$: [6]

$$z_t \in \arg \max_{z \in \mathcal{C}_t} \max_{w \in \mathcal{C}_t} w^\top \psi(s_t, z) \qquad (11)$$

---

[4]We provide an empirical comparison to this variant in Section 5.3.
[5]One additional variant is presented in Appendix B.2
[6]We additionally consider a radius of $2\beta_t$ instead of $\beta_t$ for this confidence set.

We note that, as $\mathcal{C}_t$ shrinks, so does the decision space, and with it the complexity of the optimization problem. Finally, as commonly done for linear UCB (Remark 1), we can reformulate the objective as

$$\underset{z \in \mathcal{C}_t}{\arg\max} \max_{w \in \mathcal{C}_t} w^\top \psi(s_t, z) = \underset{z \in \mathcal{C}_t}{\arg\max} \, \psi(s_t, z)^\top \hat{z}_{t-1} + \beta_t \|\psi(s_t, z)\|_{V_{t-1}^{-1}} \quad (12)$$

which allows us to optimize over just one variable. In practice, we found a simple sampling approach to be sufficient to find an approximate solution. We optimize the final objective through random shooting with a budget of $n = 128$ candidates, each of which is evaluated through a forward pass of the successor feature network $\psi$ in the BFM. Sampling from the ellipsoid $\mathcal{C}_t$ may be done efficiently by applying the push-forward $\xi \mapsto \hat{z}_{t-1} + V_{t-1}^{-1/2} \beta_t \xi$ to uniform samples from the unit ball (Barthe et al., 2005).

A practical implementation may also benefit from efficient updates to the key parameters: $\hat{z}_{t-1}$, $V_{t-1}^{-1/2}$, and $V_{t-1}^{-1}$. We do so by keeping track of the information vector $\sum_{s=1}^{t} \phi_t r_t$ and the Cholesky factor of $V_t$, i.e. $V_t = R_t^\top R_t$. This enables updates and/or recomputation of all components in $\mathcal{O}(d^2)$ (Gill et al., 1974; Seeger, 2004) making OpTI-BFM very cheap (see Appendix B.1).

**Thompson Sampling Variant**    The connection between least squares confidence sets and Bayesian linear regression is well established in contextual linear bandit literature (Kaufmann et al., 2012; Agrawal & Goyal, 2013). Concretely, we may interpret OpTI-BFM's internal components at time-step $t$ as a Gaussian posterior $\mathcal{N}(\hat{z}_{t-1}, V_{t-1}^{-1})$ over task embeddings that stem from a Bayesian linear regression with prior $\mathcal{N}(0, \frac{1}{\lambda} I_d)$ on the data $\{(\phi_i/\sigma, r_i/\sigma)\}_{i=0}^{t-1}$ (where $\sigma$ is a hyper-parameter in practice). Intuitively, this can be thought of as starting from a prior on behaviors and then refining it over time until it converges to a single behavior. Given this Bayesian interpretation, it is then natural to consider a Thompson Sampling (TS) approach, where we simply sample a behavior from the posterior, i.e. task embedding $z_t \sim \mathcal{N}(\hat{z}_t, V_t^{-1})$, which foregoes the optimization of the UCB version. We evaluate this variant extensively in Section 5.

**Non-stationary Rewards Variant**    Because of its online nature, OpTI-BFM can potentially adapt to reward functions that change over time. To this end, we consider a variant leveraging a simple idea from non-stationary bandit literature: weighing old data points less than new ones in the least squares estimator (Russac et al., 2020). Concretely, we introduce a new hyper-parameter $0 < \rho \le 1$ and weight data from past time-step $s$ at time-step $t$ with weight $\rho^{t-s}$. We evaluate OpTI-BFM in a setting with non-stationary rewards in Section 5.4.

## 5    EXPERIMENTS

The empirical evaluation is divided in several subsections, each of which will address a specific question, as their titles suggest. In the following, we first detail some evaluation choices.

**Environments**    To evaluate performance of various adaptation/inference algorithms we consider the environments Walker, Cheetah, and Quadruped from the established ExORL (Yarats et al., 2022) benchmark, with four different tasks (i.e. reward functions) each. We describe full experimental protocols in Appendix E. In all figures, error-bars and shaded regions represent min-max-intervals over 3 training seeds.

**Methods**    We choose Forward-Backward (FB) framework (Touati & Ollivier, 2021) as a state-of-the-art BFM. We adhere to the standard training and evaluation protocol for FB (Touati et al., 2023), and described it in detail in Appendix D. We apply OpTI-BFM on top of this BFM, as well as a Thompson-Sampling variant, OpTI-BFM-TS. We further consider LoLA (Sikchi et al., 2025), an approach based on policy search that was originally introduced for fast adaptation. In compliance with our online task inference setting, we initialize LoLA with an uninformed choice of $z$, and estimate on-policy returns in the standard episodic fashion, without privileged resets. Besides these three learning algorithms, we evaluate "Random" and "Oracle" baselines that serve as a lower and an upper bound on possible performance respectively. The former executes a random embedding in each step, i.e. $z_t \sim \text{Unif}(\mathcal{Z})$, whereas the latter executes the optimal policy $\pi_{z_r}$, where $z_r$ is attained by solving the linear regression problem in Eq. (4), assuming privileged access to labeled data or the reward function. We follow standard practice and approximate $z_r$ with a large budget of 50k labeled

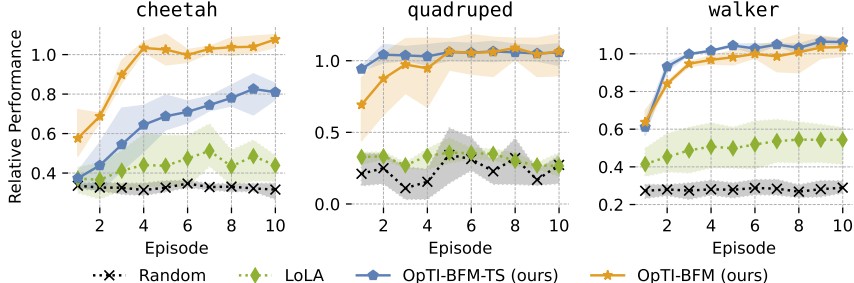

Figure 2: Mean relative performance over 10 episodes of interaction in DMC. OpTI-BFM recovers Oracle performance in 5 episodes. We report per-task absolute performance in Fig. 14 in Appendix C.

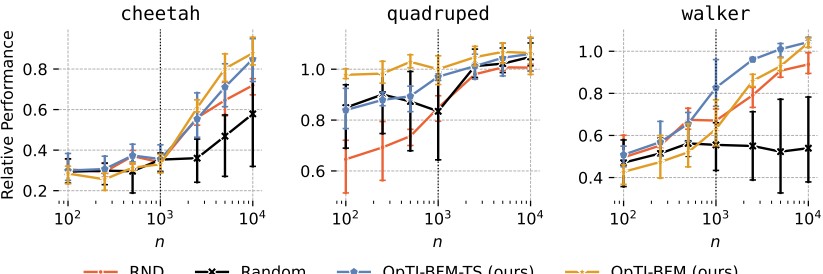

Figure 3: Relative performance after different # of environment interactions. OpTI-BFM is consistently among the top performers for all environments and time-steps. We show per task performance in Fig. 15 in Appendix C.

samples from the pre-training dataset in practice (Touati et al., 2023; Agarwal et al., 2025b). If not indicated otherwise, "relative performance" is relative to the Oracle performance.

### 5.1 HOW DOES OPTI-BFM COMPARE TO OTHER ONLINE TASK INFERENCE METHODS?

We first evaluate all methods in the online task inference setting described in Section 3.1, and report episodic returns (Eq. (5)) in Fig. 2. We find that OpTI-BFM recovers Oracle performance on all tasks within 5 episodes (5k environment steps) of interaction. We observe a significant gap between the optimistic strategy of OpTI-BFM and its TS variant in Cheetah. Nevertheless, TS remains a promising approach, as it avoids any optimization problem, and simply samples task embeddings from its current belief. Finally, we find that LoLA, which ignores the linear structure of the problem and performs blackbox policy search, makes slower progress which is better visible over 50 episodes in Fig. 16 in Appendix C. This result is consistent with existing ablations initializing LoLA to a random task embedding (see Sikchi et al. (2025), Figure 5), as is the case in our setting.

### 5.2 IS THE DATA COLLECTED ACTIVELY BY OPTI-BFM INFORMATIVE?

While the previous evaluations focus on episodic returns, or equivalently regret minimization, we now evaluate the quality of the inferred task embeddings, e.g., in the case of OpTI-BFM, how well does $\pi_{\hat{z}_n}$ perform? In practice, to ablate away any bias in $l_2$-regularized estimators, we compute the task embedding $z_n$ by minimizing the squared error in Eq. (4) over the dataset of the first $n$ observed transitions of each method, and evaluate the corresponding policy $\pi_{z_n}$ in the same environment and task. We compare with two baseline data sources: (i) random trajectories from RND (Burda et al., 2018), which represents a task-agnostic exploration approach, and (ii) the first $n$ samples from our Random baseline, which rolls out a random policy learned by the BFM. Figure 3 shows the average relative performance of $\pi_{z_n}$ in each environment. We find that the data from an actively exploring source, i.e. OpTI-BFM and RND, outperforms the passive Random approach. Furthermore, we can see that OpTI-BFM and its TS variant tend to be more data-efficient than RND, which can be traced back to task-awareness.

### 5.3 ARE FREQUENT UPDATES IMPORTANT?

The regret bound provided in Section 3.3 assumes a version of OpTI-BFM that only updates the task embedding it executes at the start of each episode. In this section we aim to experimentally

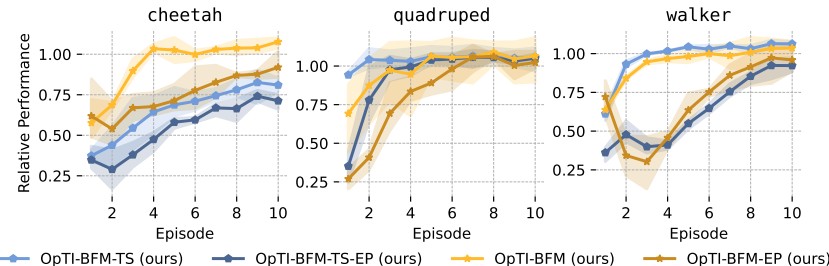

Figure 4: Relative performance of our methods, and their variants that keep task embeddings fixed for each episode (-EP). Updating the task embedding during the episode leads to faster convergence. For longer episode evaluations see Fig. 16 in Appendix C.

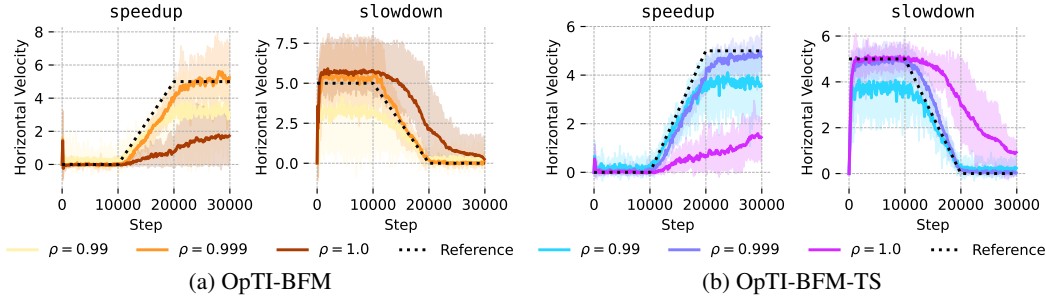

(a) OpTI-BFM        (b) OpTI-BFM-TS

Figure 5: Horizontal velocity of OpTI-BFM in our custom velocity tracking tasks in DMC Walker for different decay rates $\rho$. OpTI-BFM can adapt to non-stationary reward functions when decaying the weight of old observations.

quantify this gap in theory and practice by evaluating this episodic version of OpTI-BFM. Fig. 4 shows that changing the latent on an episode-level leads to slower improvement than changing every step. The episodic versions do reach equal performance eventually as shown in Fig. 16 in Appendix C. Intuitively, this faster improvement may be explained by the fact that the default version of OpTI-BFM can adjust faster to new information.

### 5.4 CAN OpTI-BFM ADAPT TO NON-STATIONARY REWARDS?

It is conceivable that OpTI-BFM can adapt to reward functions that are changing over time because of its online nature. To investigate whether this is possible, we introduce two new tasks for the Walker environment that closely resemble the `walk` and `run` tasks but change the velocity target over time: `speedup` and `slowdown` increase and decrease the velocity target respectively after an initial "burn-in" phase. We evaluate OpTI-BFM in a single, 30k-step, episode in Figure 5, which shows that a direct application of OpTI-BFM ($\rho = 1.0$) struggles to adapt to the changing reward function after converging to a fixed one in the first 10k steps. As soon as $\rho$ is reduced, OpTI-BFM tracks the velocity target more accurately. Moreover, we observe that, if $\rho$ is too small, the uncertainty is not reduced quickly enough, and the agent can converge to suboptimal behavior.

### 5.5 CAN OpTI-BFM WARM-START FROM LABELED DATA?

In settings that provide a small amount of labeled data $\mathcal{D} = \{(s_i, r_i)\}_{i=1}^{n}$ from the beginning (Sikchi et al., 2025), OpTI-BFM may be warm-started by updating the least squares estimator $n$-times with the given data. As shown in Fig. 6, when doing so on high-quality i.i.d. data from the training dataset, the performance of OpTI-BFM quickly improves, showing that OpTI-BFM can additionally be seen an *extension* of the traditional task inference of BFMs.

## 6 RELATED WORK

**Behavior Foundation Models** This work builds upon USFs (Dayan, 1993; Barreto et al., 2017; Ma et al., 2020), and specifically on BFMs that utilize SFs to train a parameterized policy and the corresponding parameterized SFs. A common approach for learning SF-based BFMs is to

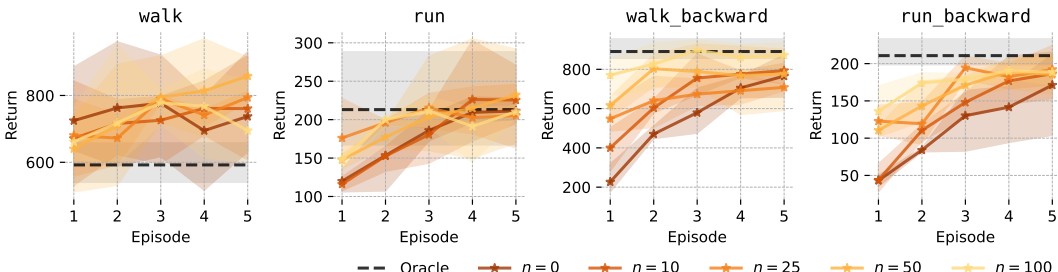

Figure 6: Return of OpTI-BFM in DMC Cheetah when warm-starting with $n$ i.i.d. labeled states from the training dataset. Initial performance increases quickly as $n$ grows. We report full results in Fig. 17 in Appendix C.

.

approximate of successor measures (Blier et al., 2021; Touati & Ollivier, 2021; Agarwal et al., 2025b), but other options spanning from spectral decomposition of random visitations (Wu et al., 2018) to implicit value learning (Park et al., 2024b) also exist. In this regard, we refer the reader to Agarwal et al. (2025a) for a broader overview of BFMs and unsupervised RL. Among these approaches, the Forward Backward (FB) framework (Touati & Ollivier, 2021) has risen to prominence, as it achieves state-of-the-art performance in standard continuous control tasks (Touati et al., 2023), while being scalable to humanoid control (Tirinzoni et al., 2025). Recent work has explored various applications of FB's zero-shot capabilities, such as imitation learning (Pirotta et al., 2023), epistemic exploration (Urpí et al., 2025), and constrained RL (Hugessen et al., 2025). Among related works, our method is most closely related to Sikchi et al. (2025), which however focuses on a different problem: fine-tuning a zero-shot policy, which is already initialized through standard offline task inference. On the other hand, our goal is to find a strong policy purely through environment interaction, without relying on preexisting labeled data. The two approaches might nevertheless be combined.

**Reward Learning**   The task inference goal of OpTI-BFM is fundamentally connected to the reward learning problem in RL. Among the many instantiations, rewards may be inferred from scalar evaluations (Knox & Stone, 2009; MacGlashan et al., 2017), preferences (Fürnkranz et al., 2012; Christiano et al., 2017), or from other types of feedback (Jeon et al., 2020). A powerful extension to reward learning explores active strategies for query selection, largely adopting a Bayesian perspective (Bıyık et al., 2020; Wilde et al., 2020). While none of these works are aimed at BFMs, several are close in spirit: for instance, Lindner et al. (2021) propose an information-directed method to achieve a similar goal to ours: inferring the task in a way which does not necessarily reduce the model error, but quickly produces an optimal policy.

**Linear Bandits**   Linear Contextual Bandits are a well-studied problem in bandit literature (Abe & Long, 1999; Dani et al., 2008; Abbasi-Yadkori et al., 2011; Kaufmann et al., 2012; Agrawal & Goyal, 2013). This is influential to this work in two ways. First, OpTI-BFM is inspired by UCB methods in this setting (Dani et al., 2008; Abbasi-Yadkori et al., 2011). Second, we utilize guarantees and ideas from this literature (Lattimore & Szepesvári, 2020) to produce a regret bound for OpTI-BFM in an episodic setting.

## 7   CONCLUSION

By drawing a connection between practical work in scalable Behavior Foundation Models and rigorous algorithms for linear optimization, we proposed an algorithm that may efficiently infer the task at hand and retrieve a well-performing policy in high-dimensional, complex environments. By minimizing the number of reward evaluations necessary for task inference, this algorithm can enable BFM to be applied beyond domains in which rewards are readily available, for instance when learning directly from pixels.

OpTI-BFM is designed to perform task inference in a minimal number of episodes: this is mainly possible as the search space is minimal. While updating the task embedding alone enables great sample efficiency, fine-tuning additional components of the BFM may provide even better performance in the long run (Sikchi et al., 2025). Moreover, our theoretical guarantees only cover slower, episode-level

updates: extending these results to (empirically stronger) per-step represents an important direction for future formal works.

While OpTI-BFM builds upon linearity between features and rewards, it does not make any significant assumption on the structure of the feature space $\mathcal{Z}$: understanding the properties of its elements both formally and practically constitutes an important avenue for future work. BFMs have demonstrated scalability up to complex domains, while maintaining structured properties (linearity!) in an embedding space: we believe that this is a ripe field for application of more theoretically principled approaches, which may in turn be impactful beyond regular domains.

## ACKNOWLEDGEMENTS

We thank Núria Armengol Urpí and Georg Martius for their help throughout the project. Marco Bagatella is supported by the Max Planck ETH Center for Learning Systems. We acknowledge the support from the Swiss National Science Foundation under NCCR Automation, grant agreement 51NF40 180545.

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

## A  PROOFS

This section describes the proof for Proposition 1. It closely follows Chapters 19 and 20 from Lattimore & Szepesvári (2020). We proceed as follows:

- We reintroduce the setting with some more convenient notation in Appendix A.1.

- We show that running linear regression on feature-reward pairs is at least as efficient as running linear regression on SF-return pairs (Appendix A.2)

- We cite a result that grants the optimal choice of $\beta$s in Appendix A.3.

- We proceed with the standard linear contextual bandit regret bound proof with confidence sets based on feature-reward pairs (Appendix A.4).

### A.1  SETTING

Throughout this section we will use the double subscript $k, t$ to denote the $t$-th time-step in the $k$ episode. Since we will analyze a version of OpTI-BFM that only switches task embedding at the start of each episode, and since we know that the best task embedding is always $z_r$ (Eq. 4), we define $z_1, \ldots, z_n$ be the chosen task embeddings for $n$ episodes. Let $\phi_{k,t} = \phi(s_{k,t})$ be the feature observed in step $t$ of episode $k$. Note that this quantity is a random variable; throughout this section we use a simple expectation $\mathbb{E}[\cdot]$ to indicate the expectation over initial states, action distributions of policies, MDP dynamics, and decisions of the algorithm. We will still use $\mathbb{E}\left[\cdot \mid \pi, s\right]$ to denote the expectation over rolling out policy $\pi$ from state $s$.

#### A.1.1  ASSUMPTIONS

For convenience, we repeat our assumptions here with the new notation

(A1) Perfect USF for our setting: for every $(s, a, z) \in \mathcal{S} \times \mathcal{A} \times \mathcal{Z}$ we have

$$\psi(s_0, a_0, z) = \sum_{t=0}^{H-1} \gamma^t \mathbb{E}\left[\phi(s_t) \mid \pi_z, s_0, a_0\right] \tag{13}$$

and

$$\pi_z(a|s) > 0 \implies a \in \underset{a \in \mathcal{A}}{\arg\max} \, \psi(s, a, z)^\top z. \tag{14}$$

(A2) Linear Reward: $r$ is in the span of features $\phi$ up to i.i.d. mean-zero $\sigma$-subgaussian noise $\eta_{k,t}$, i.e.

$$r_{k,t} = \phi(s_{k,t})^\top z_r + \eta_{k,t} \tag{15}$$

(A3) Optimization Oracle: the OpTI-BFM objective in Eq. (10) can be computed exactly.

(A4) Bounded norms: $\|z_r\|_2 \le S$ and $\|\phi(\cdot)\|_2 \le L$ for some $S > 0$ and $L > 0$.

#### A.1.2  REGRET

We also repeat the definitions of discounted return of the $k$-th episode (Eq. (5)) and expected regret (Eq. (6)) for consistent notation.

$$\hat{G}_k = \sum_{t=0}^{H-1} \gamma^t r_{k,t} \tag{16}$$

$$R_n = \mathbb{E}\left[\sum_{k=1}^{n} \hat{G}_k^\star - \hat{G}_k\right], \tag{17}$$

where $\hat{G}_k^\star$ is the discounted return achieved by $\pi_{z_r}$ which is optimal for the the reward $r(\cdot) = \phi(\cdot)^\top z_r$ under A1. We now also define the expected discounted return of episode $k$ for task embedding $z$ as

$$G_{k,z} = \mathbb{E}\left[\hat{G}_k \mid \pi_z, s_{k,0}\right]. \tag{18}$$

Notice that this implies that $\mathbb{E}[\hat{G}_k^\star] = G_{k,z_r}$ and that $R_n = \mathbb{E}[\sum_{k=1}^n G_{k,z^\star} - G_{k,z_k}]$, the latter of which is the term we want to bound at the end of this section. We may now show that the expected discounted return is linear in the SFs:

$$G_{k,z} = \mathbb{E}\left[\hat{G}_k \mid \pi_z, s_{k,0}\right] \tag{19}$$

$$= \mathbb{E}\left[\sum_{t=0}^{H-1} \gamma^t r_{k,t} \mid \pi_z, s_{k,0}\right] \tag{20}$$

$$= \mathbb{E}\left[\sum_{t=0}^{H-1} \gamma^t \langle z_r, \phi(s_{k,t})\rangle \mid \pi_z, s_{k,0}\right] \tag{21}$$

$$= \left\langle z_r, \mathbb{E}\left[\sum_{t=0}^{H-1} \gamma^t \phi(s_{k,t}) \mid \pi_z, s_{k,0}\right]\right\rangle \tag{22}$$

$$\overset{A1}{=} \langle z_r, \psi(s_{k,0}, z)\rangle \tag{23}$$

Thus, our setting resembles a contextual bandit with context $s_0 \sim \mu_0$, action space $\mathcal{Z}$ and an expected payoff that is linear in $\psi(s_0, z)$, with the one notable difference being that more information is available on the pay-off feedback: it is not available in terms of return $\hat{G}_k$ alone, but additionally as a sequence of features-reward pairs $(\phi_{k,t}, r_{k,t})_{t=0}^{H-1}$.

### A.1.3 OpTI-BFM

Let us finally reintroduce the least squares estimator with the double subscript notation:

$$\hat{z}_n = V_n^{-1} \sum_{k=1}^n \sum_{t=0}^{H-1} \phi_{k,t} r_{k,t} \quad \text{where} \quad V_n = \lambda I + \sum_{k=1}^n \sum_{t=0}^{H-1} \phi_{k,t} \phi_{k,t}^\top. \tag{24}$$

From this we then redefine the confidence sets at the start of episode $k$ as

$$\mathcal{C}_k = \{z \in \mathbb{R}^d : \|z - \hat{z}_{k-1}\|_{V_{k-1}} \leq \beta_k\}. \tag{25}$$

Defining the UCB operator as

$$\mathrm{UCB}_k(z) = \max_{w \in \mathcal{C}_k} \langle \psi(s_{k,0}, z), w\rangle, \tag{26}$$

then OpTI-BFM picks $z_k = \arg\max_{\mathcal{Z}} \mathrm{UCB}_k(z)$ for the $k$-th episode, which we assume can be attained exactly in A3.

## A.2 RETURN-LEVEL VS. REWARD-LEVEL FEEDBACK

In this section we show a few properties conveying the intuitive fact that, if $z_r$ is estimated from richer, reward-level feedback $(\phi_{k,t}, r_{k,t})$, confidence sets may be tigher than those produced by return-level feedback $(\sum_{t=0}^{H-1} \gamma^t \phi_{k,t}, \sum_{t=0}^{H-1} \gamma^t r_{k,t})$, which can be though of as an aggregation of $H$ datapoints of the former (up to expectation over dynamics).

For this analysis we define the *empirical* SF of episode $k$ as

$$\tilde{\psi}_k = \sum_{t=0}^{H-1} \gamma^t \phi_{k,t} \quad \text{and} \quad W_n = \lambda I + \sum_{k=1}^n \tilde{\psi}_k \tilde{\psi}_k^\top. \tag{27}$$

Note that we have $\mathbb{E}\left[\tilde{\psi}_k \mid \pi_{z_k}, s_{k,0}\right] = \psi(s_{k,0}, z_k)$. We further define

$$A_k = \sum_{t=0}^{H-1} \phi_{k,t} \phi_{k,t}^\top, \tag{28}$$

so that we have $V_n = \lambda I + \sum_{k=1}^n A_k$. Finally, the following constant [7] will become useful later:

$$c_H = \sum_{t=0}^{H-1} \gamma^{2t} = \frac{1 - \gamma^{2H}}{1 - \gamma^2}. \tag{29}$$

We now show that the precision matrices obtained through reward-level feedback ($V_n$) grow at least as fast as those obtained through return-level feedback ($W_n$) over the course of one episode.

---

[7]Note that $c_H \geq 1$ for $H > 0$.

**Proposition 2.** *We have, in Loewner order,*

$$\tilde{\psi}_k \tilde{\psi}_k^\top \preccurlyeq c_H A_k. \tag{30}$$

*Proof.* This is a direct application of a Cauchy-Schwarz inequality: for any $x$,

$$x^\top \tilde{\psi}_k \tilde{\psi}_k^\top x = (x^\top \tilde{\psi}_k)^2 = \left( \sum_{t=0}^{H-1} \gamma^t x^\top \phi_{k,t} \right)^2 \leq \left( \sum_{t=0}^{H-1} \gamma^{2t} \right) \left( \sum_{t=0}^{H-1} (x^\top \phi_{k,t})^2 \right) = c_H x^\top A_k x$$

$\square$

From this, it immediately follows $V_n$ grows at least as fast as $W_n$ over all episodes.

**Proposition 3.** *We have, in Loewner order,*

$$V_n \succcurlyeq \frac{1}{c_H} W_n \tag{31}$$

*Proof.* Using Proposition 2, we know that

$$A_k \succcurlyeq \frac{1}{c_H} \tilde{\psi}_k \tilde{\psi}_k^\top. \tag{32}$$

Summing over episodes on both sides, we get

$$\sum_{k=1}^{n} A_k \succcurlyeq \frac{1}{c_H} \sum_{k=1}^{n} \tilde{\psi}_k \tilde{\psi}_k^\top. \tag{33}$$

By adding $\lambda I$ on both sides, we can write

$$\lambda I + \sum_{k=1}^{n} A_k \succcurlyeq \lambda I + \frac{1}{c_H} \sum_{k=1}^{n} \tilde{\psi}_k \tilde{\psi}_k^\top. \tag{34}$$

On the left, we recognize

$$\lambda I + \sum_{k=1}^{n} A_k = V_n, \tag{35}$$

while on the right we have

$$\lambda I + \frac{1}{c_H} \sum_{k=1}^{n} \tilde{\psi}_k \tilde{\psi}_k^\top = \frac{1}{c_H} \left( \lambda I + \sum_{k=1}^{n} \tilde{\psi}_k \tilde{\psi}_k^\top \right) + \left( 1 - \frac{1}{c_H} \right) \lambda I = \frac{1}{c_H} W_n + \left( 1 - \frac{1}{c_H} \right) \lambda I,$$

which in turn yields

$$V_n \succcurlyeq \frac{1}{c_H} W_n + \left( 1 - \frac{1}{c_H} \right) \lambda I \succcurlyeq \frac{1}{c_H} W_n, \tag{36}$$

where the last step follows from the fact that $c_H \geq 1$. $\square$

We now derive a useful property from this result, that will later help us to apply the well-known elliptical potential Lemma. Intuitively it shows that reward-level feedback leads to confidence ellipsoids at least as tight as those of return-level feedback.

**Proposition 4.** *We have for any $x \in \mathbb{R}^d$*

$$\|x\|_{V_n^{-1}} \leq \sqrt{c_H} \|x\|_{W_n^{-1}} \tag{37}$$

*Proof.* By Proposition 3,

$$V_n \succcurlyeq \frac{1}{c_H} W_n, \tag{38}$$

where both matrices are positive definite by construction. As the inverse thus reverses order,

$$V_n^{-1} \preccurlyeq c_H W_n^{-1}. \tag{39}$$

As a consequence, one can show that, for any $x$,

$$\|x\|_{V_n^{-1}} = \sqrt{x^\top V_n^{-1} x} \leq \sqrt{c_H x^\top W_n^{-1} x} = \sqrt{c_H} \|x\|_{W_n^{-1}}. \tag{40}$$

$\square$

## A.3 Confidence Ellipsoids

We will now take take the chance to cite a standard result for optimal choices of radii $\beta_k$ which guarantee that the true parameter $z_r$ is contained in confidence sets (Eq. (9))in each time-step with high probability.

**Theorem 1.** *(Adapted from Abbasi-Yadkori et al. (2011) Theorem 2)*
*Given confidence sets $\mathcal{C}_k$ as defined in Eq. (25), and if rewards are in the span of $\phi$ up to $\sigma$-subgaussian zero-mean noise (A2), then for any $\delta \in (0, 1)$ the following holds:* $\Pr[\exists k \in \mathbb{N}^+ | z_r \notin \mathcal{C}_k] \leq \delta$ *with*

$$\beta_k = \sqrt{\lambda}S + \sigma\sqrt{\log\left(\frac{\det V_{k-1}}{\lambda^d}\right) + 2\log\left(\frac{1}{\delta}\right)} \tag{41}$$

This is the classic uniform-in-time self-normalized bound based on supermartingales. Notice that the theorem makes no assumptions about the action selection process nor the process that controls the state-changes, except that they are measurable in the natural filtration of the corresponding time-step, which is the case for our decision rule (OpTI-BFM) and our episodic MDP setting. We remark that, since by definition $V_t \succcurlyeq V_s$ for all $t \geq s$, we have that $\beta_t \geq \beta_s$ as well.

## A.4 Regret Bound

We can now prove a formal version of Proposition 1.

**Proposition 5.** *Under assumptions A1-A4, with choices for $\beta_k$ as in Theorem 1, if OpTI-BFM is applied at the beginning of each episode and selects the task embedding $z_k = \arg\max_{\mathcal{Z}} UCB_k(z)$ for the $k$-th episode, it incurs an expected regret of $\tilde{\mathcal{O}}(d\sqrt{n})$.*

*Proof.* We start off by investigating the event $\mathcal{E}$ that $z_r \in \mathcal{C}_k \, \forall k \in \{1, \ldots, n\}$, i.e., $z_r$ is in all confidence sets, which occurs with probability at least $1 - \delta$ according to Theorem 1.

Since we will refer to both empirical and "expected" SFs ($\tilde{\psi}$ and $\psi$, respectively), we introduce the natural filtration $\mathcal{F}_{k-1}$ generated by the history up to the start of episode $k$, in particular, $s_{k,0}, z_k, V_{k-1}, W_{k-1}$ are $\mathcal{F}_{k-1}$-measurable. Intuitively, conditioning on $\mathcal{F}_{k-1}$ allows us to limit randomness to within the current episode. Crucial for us will be,

$$\psi(s_{k,0}, z_k) = \mathbb{E}\left[\tilde{\psi}_k \mid \mathcal{F}_{k-1}\right] \tag{42}$$

We will proceed as in the standard regret bound for linear contextual bandits (Lattimore & Szepesvári, 2020) Let $\tilde{z}_k = \arg\max_{w \in \mathcal{C}_k}\langle\psi(s_{k,0}, z_k), w\rangle$ be the task embedding that maximizes the UCB. We then have that

$$\langle\psi(s_{k,0}, z_r), z_r\rangle \leq UCB_k(z_r) \leq UCB_k(z_k) = \langle\psi(s_{k,0}, z_k), \tilde{z}_k\rangle. \tag{43}$$

We can therefore bound the instantaneous regret $e_k$ by

$$e_k = G_{k,z_r} - G_{k,z_k} \tag{44}$$

$$= \langle\psi(s_{k,0}, z_r), z_r\rangle - \langle\psi(s_{k,0}, z_k), z_r\rangle \tag{45}$$

$$\leq \langle\psi(s_{k,0}, z_k), \tilde{z}_k\rangle - \langle\psi(s_{k,0}, z_k), z_r\rangle \tag{46}$$

$$= \langle\psi(s_{k,0}, z_k), \tilde{z}_k - z_r\rangle. \tag{47}$$

Conditioning on previous events ($\mathcal{F}_{k-1}$), and applying the Cauchy-Schwarz inequality pointwise, we have

$$\mathbb{E}[e_k \mid \mathcal{F}_{k-1}] \leq \mathbb{E}[\langle\psi(s_{k,0}, z_k), \tilde{z}_k - z_r\rangle \mid \mathcal{F}_{k-1}] \tag{48}$$

$$= \mathbb{E}\left[\langle\tilde{\psi}_k, \tilde{z}_k - z_r\rangle \mid \mathcal{F}_{k-1}\right] \tag{49}$$

$$\leq \mathbb{E}\left[\left\|\tilde{\psi}_k\right\|_{V_{k-1}^{-1}} \|\tilde{z}_k - z_r\|_{V_{k-1}} \mid \mathcal{F}_{k-1}\right]. \tag{50}$$

Since we are in event $\mathcal{E}$ we have $z_r \in \mathcal{C}_k$, furthermore, $\tilde{z}_k \in \mathcal{C}_k$ by definition. So, using the definition of the confidence set Eq. (25), we have

$$\|\tilde{z}_k - z_r\|_{V_{k-1}} \leq 2\beta_k \leq 2\beta_n. \tag{51}$$

This is where this proof diverges from the standard regret bound: note that $V_{k-1}$ is a sum of features $\phi$ *and not* empirical SFs $\tilde{\psi}$, making immediate application of the elliptical potential lemma non-trivial. We can however recall the comparison of least square estimation with reward-feedback and return-feedback from Appendix A.2. In particular, one can apply Proposition 4 to upper-bound the $V$-norm with the $W$-norm that depends on the empirical SFs:

$$\left\|\tilde{\psi}_k\right\|_{V_{k-1}^{-1}} \leq \sqrt{c_H} \left\|\tilde{\psi}_k\right\|_{W_{k-1}^{-1}}. \tag{52}$$

Combining these two bounds results in

$$\mathbb{E}\left[e_k \mid \mathcal{F}_{k-1}\right] \leq \mathbb{E}\left[2\beta_n \sqrt{c_H} \left\|\tilde{\psi}_k\right\|_{W_{k-1}^{-1}} \Big| \mathcal{F}_{k-1}\right]. \tag{53}$$

Recalling that, by assumption A4, $\left\|\tilde{\psi}_k\right\|_2 \leq \frac{L(1-\gamma^H)}{1-\gamma}$ and

$$V_{k-1} = \lambda I + \sum_{i=1}^{k} \sum_{t=0}^{H-1} \phi_{i,t}\phi_{i,t}^\top \succcurlyeq \lambda I \tag{54}$$

so $\|u\| \leq \lambda^{-1/2} \|u\|_{V_{k-1}}$, we can write that, pointwise for all $k$,

$$\left|\left\langle \tilde{\psi}_k, \tilde{z}_k - z_r \right\rangle\right| \leq \min\left\{\frac{2\beta_n}{\sqrt{\lambda}} \left\|\tilde{\psi}_k\right\|, \, 2\beta_n \sqrt{c_H} \left\|\tilde{\psi}_k\right\|_{W_{k-1}^{-1}}\right\} \tag{55}$$

$$\leq 2\beta_n \min\left\{\underbrace{\frac{L(1-\gamma^H)}{\sqrt{\lambda}(1-\gamma)}}_{=:B}, \, \sqrt{c_H} \left\|\tilde{\psi}_k\right\|_{W_{k-1}^{-1}}\right\}. \tag{56}$$

Combining the two bounds inside the expectation (as they hold pointwise), we obtain

$$\mathbb{E}\left[e_k \mid \mathcal{F}_{k-1}\right] \leq \mathbb{E}\left[2\beta_n \min\left\{B, \sqrt{c_H} \left\|\tilde{\psi}_k\right\|_{W_{k-1}^{-1}}\right\} \Big| \mathcal{F}_{k-1}\right]. \tag{57}$$

Using the fact that $\min\{a, bx\} \leq \max\{a, b\}\min\{1, x\}$ for $a, b > 0$, we can then take some constants out of the expectation:

$$\mathbb{E}\left[e_k \mid \mathcal{F}_{k-1}\right] \leq \underbrace{\max\{B, \sqrt{c_H}\}}_{=:\alpha} \mathbb{E}\left[2\beta_n \min\left\{1, \left\|\tilde{\psi}_k\right\|_{W_{k-1}^{-1}}\right\} \Big| \mathcal{F}_{k-1}\right]. \tag{58}$$

We now sum over episodes. To do this, we use the tower rule on $e_k$, i.e., $\mathbb{E}[e_k] = \mathbb{E}[\mathbb{E}[e_k \mid \mathcal{F}_{k-1}]]$.

$$R_n = \mathbb{E}\left[\sum_{k=1}^{n} e_k\right] = \sum_{k=1}^{n} \mathbb{E}[e_k] = \sum_{k=1}^{n} \mathbb{E}\left[\mathbb{E}[e_k \mid \mathcal{F}_{k-1}]\right] \tag{59}$$

Applying our (pointwise) bound on $\mathbb{E}[e_k \mid \mathcal{F}_{k-1}]$ and the tower rule again, we get.

$$R_n = \sum_{k=1}^{n} \mathbb{E}\left[\mathbb{E}[e_k \mid \mathcal{F}_{k-1}]\right] \leq 2\alpha \sum_{k=1}^{n} \mathbb{E}\left[\mathbb{E}\left[\beta_n \min\left\{1, \left\|\tilde{\psi}_k\right\|_{W_{k-1}^{-1}}\right\} \Big| \mathcal{F}_{k-1}\right]\right] \tag{60}$$

$$= 2\alpha \sum_{k=1}^{n} \mathbb{E}\left[\beta_n \min\left\{1, \left\|\tilde{\psi}_k\right\|_{W_{k-1}^{-1}}\right\}\right]. \tag{61}$$

Using linearity of expectation and then applying the Cauchy-Schwarz inequality over episodes (pointwise), we can prepare the sum to the application of the elliptical potential Lemma on $(\tilde{\psi}_k)_{k=1}^{n}$

and $(W_k)_{k=1}^n$ (pathwise):

$$R_n \leq 2\alpha \, \mathbb{E}\left[\beta_n \sum_{k=1}^n \min\left\{1, \left\|\tilde{\psi}_k\right\|_{W_{k-1}^{-1}}\right\}\right] \tag{62}$$

$$\leq 2\alpha \, \mathbb{E}\left[\beta_n \sqrt{n \cdot \sum_{k=1}^n \min\left\{1, \left\|\tilde{\psi}_k\right\|_{W_{k-1}^{-1}}^2\right\}}\right] \tag{63}$$

$$\leq 2\alpha \, \mathbb{E}\left[\beta_n \sqrt{2n \cdot \log\left(\frac{\det W_n}{\lambda^d}\right)}\right]. \qquad\qquad \text{Lemma 1} \tag{64}$$

Inserting our choice for $\beta_n$ we obtain

$$R_n \leq 2\alpha \, \mathbb{E}\left[\left(\sqrt{\lambda}S + \sigma\left(\sqrt{\log\left(\frac{\det V_{n-1}}{\lambda^d}\right) + 2\log(1/\delta)}\right)\right)\sqrt{2n \cdot \log\left(\frac{\det W_n}{\lambda^d}\right)}\right] \tag{65}$$

We recall that, so far, we investigated the expected regret conditional on the event $\mathcal{E}$ that $z_r \in \mathcal{C}_k \, \forall k \in \{1, \ldots, n\}$, which occurs with probability at least $1 - \delta$ according to Theorem 1. We now choose $\delta = 1/n$. Using Proposition 6 to upper-bound the determinants (pointwise), and recalling the fact that $\left\|\tilde{\psi}_k\right\|_2 \leq \frac{L(1-\gamma^H)}{1-\gamma}$ (by A4), we obtain

$$R_n \leq 2\alpha\left(\sqrt{\lambda}S + \sigma\left(\sqrt{d\log\left(\frac{d\lambda + nHL^2}{d\lambda}\right) + 2\log n}\right)\right)\sqrt{2nd\log\left(\frac{d\lambda + n\frac{L^2(1-\gamma^H)^2}{(1-\gamma)^2}}{d\lambda}\right)}$$

$$\leq \tilde{\mathcal{O}}\left(d\sqrt{n}\right). \tag{66}$$

We finally investigate the complementary event $\mathcal{E}^C$. We remark that the discounted return in each episode can be bounded by constants using the Cauchy-Schwarz inequality:

$$|G_{k,z}| = \left|\mathbb{E}\left[\sum_{t=0}^{H-1} \gamma^t \phi_{k,t}^\top z_r \,\middle|\, \pi_{z_k}, s_{k,0}\right]\right| \leq LS \sum_{t=0}^{H-1} \gamma^t \leq \frac{LS}{1-\gamma}. \tag{67}$$

It thus holds that $R_n \leq n \cdot \frac{2LS}{1-\gamma}$ under $\mathcal{E}^C$.

To combine the analysis for both events, let $\hat{R}_n = \sum_{k=1}^n e_k$ be the empirical regret. By the law of total probability, we have that

$$R_n = \mathbb{E}[\hat{R}_n] = \Pr[\mathcal{E}] \cdot \mathbb{E}\left[\hat{R}_n \,\middle|\, \mathcal{E}\right] + \Pr[\mathcal{E}^C] \cdot \mathbb{E}\left[\hat{R}_n \,\middle|\, \mathcal{E}^C\right] \tag{68}$$

$$\leq 1 \cdot \tilde{\mathcal{O}}\left(d\sqrt{n}\right) + \frac{1}{n} \cdot n \cdot \frac{2LS}{1-\gamma} \tag{69}$$

$$\leq \tilde{\mathcal{O}}\left(d\sqrt{n}\right), \tag{70}$$

which concludes the proof. $\qquad\qquad\qquad\qquad\qquad\qquad\qquad\qquad\qquad\qquad\qquad\qquad\qquad\square$

### A.5 REGRET BOUND UNDER WEAKER ASSUMPTIONS

Our empirical evaluations are performed in a setting which violates two assumptions described in A.1.1: perfect USF estimation (A1) and linearity of rewards (A2). This section provides a study of the algorithm's behavior under these violations taking some ideas from existing theory of misspecified bandits (Ghosh et al., 2017; Bogunovic & Krause, 2021; Lattimore & Szepesvári, 2020).

We start by replacing A1 and A2 by weaker assumptions that quantify the mismatch between $\phi$ and $\psi$ and between $r$ and its projection $z_r^\top \psi$ respectively:

(A1') For every $(s, a, z) \in \mathcal{S} \times \mathcal{A} \times \mathcal{Z}$ we have

$$\left\| \psi(s, a, z) - \sum_{t=0}^{H-1} \gamma^t \mathbb{E} \left[ \phi(s_t) \mid \pi_z, s, a \right] \right\|_2 \leq \xi \tag{71}$$

and

$$\pi_z(a|s) > 0 \implies a \in \underset{a \in \mathcal{A}}{\arg\max} \, \psi(s, a, z)^\top z. \tag{72}$$

(A2') For reward $r$ we have for each state $s_{k,t}$

$$\mathbb{E}[\|r(s_{k,t}) - \phi(s_{k,t})^\top z_r\|] \leq \zeta \tag{73}$$

where the expectation is over i.i.d. mean-zero $\sigma$-subgaussian noise $\eta_{k,t}$.

We may thus write $\psi$ as a biased version of the true $\psi^\star$

$$\psi(s, a, z) + \Delta_\psi(s, a, z) = \psi^*(s, a, z), \tag{74}$$

and, similarly, rewrite rewards as

$$r_{k,t} = \phi(s_{k,t})^\top z_r + \Delta_r(s_{k,t}) + \eta_{k,t}. \tag{75}$$

We can then establish a uniform-in-time bound as in Theorem 1 by inflating $\beta$ to account for the newly introduced bias in the reward. Considering the distance between the current mean estimate $\hat{z}_{t-1}$ and $z_r$, we can decompose this into

$$\|z_r - \hat{z}_{t-1}\|_{V_{t-1}} = \left\| z_r - \sum_{i=1}^{t-1} \phi_i \phi_i^\top z_r - V_{t-1}^{-1} \sum_{i=1}^{t-1} \phi_i \eta_i - V_{t-1}^{-1} \sum_{i=1}^{t-1} \phi_i \Delta_r(s_i) \right\|_{V_{t-1}} \tag{76}$$

$$= \left\| z_r - V_{t-1} z_r + \lambda z_r - V_{t-1}^{-1} \sum_{i=1}^{t-1} \phi_i \eta_i - V_{t-1}^{-1} \sum_{i=1}^{t-1} \phi_i \Delta_r(s_i) \right\|_{V_{t-1}} \tag{77}$$

$$= \left\| z_r - z_r - V_{t-1}^{-1} \left( \lambda z_r + \sum_{i=1}^{t-1} \phi_i \eta_i + \sum_{i=1}^{t-1} \phi_i \Delta_r(s_i) \right) \right\|_{V_{t-1}} \tag{78}$$

$$= \left\| \lambda z_r + \sum_{i=1}^{t-1} \phi_i \eta_i + \sum_{i=1}^{t-1} \phi_i \Delta_r(s_i) \right\|_{V_{t-1}^{-1}} \tag{79}$$

$$\leq \left\| \lambda z_r + \sum_{i=1}^{t-1} \phi_i \eta_i \right\|_{V_{t-1}^{-1}} + \left\| \sum_{i=1}^{t-1} \phi_i \Delta_r(s_i) \right\|_{V_{t-1}^{-1}} \tag{80}$$

The first term is covered by Theorem 1, while the new bias term can be bounded through similar techniques to those used in the main proof:

$$\left\| \sum_{i=1}^{t-1} \phi_i \Delta_r(s_i) \right\|_{V_{t-1}^{-1}} \leq \sum_{i=1}^{t-1} |\Delta_r(s_i)| \, \|\phi_i\|_{V_{t-1}^{-1}} \qquad \|\cdot\|_{V_{t-1}^{-1}} \text{ properties} \qquad (81)$$

$$\leq \zeta \sum_{i=1}^{t-1} \|\phi_i\|_{V_{t-1}^{-1}} \qquad \text{A2'} \qquad (82)$$

$$\leq \zeta \sum_{i=1}^{t-1} \|\phi_i\|_{V_{i-1}^{-1}} \qquad V_{i-1} \preccurlyeq V_{t-1} \qquad (83)$$

$$\leq \zeta \sqrt{t \sum_{i=1}^{t-1} \|\phi_i\|_{V_{i-1}^{-1}}^2} \qquad \text{Cauchy-Schwarz} \qquad (84)$$

$$\leq \zeta \sqrt{t \sum_{i=1}^{t-1} \min\{\tfrac{L^2}{\lambda}, \|\phi_i\|_{V_{i-1}^{-1}}^2\}} \qquad (85)$$

$$\leq \frac{L\zeta}{\sqrt{\lambda}} \sqrt{t \sum_{i=1}^{t-1} \min\{1, \|\phi_i\|_{V_{i-1}^{-1}}^2\}} \qquad (86)$$

$$\leq \frac{\sqrt{2}L\zeta}{\sqrt{\lambda}} \sqrt{t \log\left(\frac{\det V_t}{\det V_0}\right)} \qquad \text{Lemma 1} \qquad (87)$$

As a result, we can establish a uniform-in-time guarantee if $\beta$ is inflated as

$$\beta'_t = \beta_t + \frac{\sqrt{2}L\zeta}{\sqrt{\lambda}} \sqrt{t \log\left(\frac{\det V_t}{\det V_0}\right)}. \qquad (88)$$

Continuing to build the regret bound, we first notice that the mismatch also impacts the discounted return:

$$G_{k,z} = \mathbb{E}\left[\hat{G}_k \mid \pi_z, s_{k,0}\right] \qquad (89)$$

$$= \mathbb{E}\left[\sum_{t=0}^{H-1} \gamma^t r_{k,t} \mid \pi_z, s_{k,0}\right] \qquad (90)$$

$$\overset{A2'}{=} \mathbb{E}\left[\sum_{t=0}^{H-1} \gamma^t \langle z_r, \phi(s_{k,t})\rangle + \Delta_r(s_{k,t}) \mid \pi_z, s_{k,0}\right] \qquad (91)$$

$$= \left\langle z_r, \mathbb{E}\left[\sum_{t=0}^{H-1} \gamma^t \phi(s_{k,t}) \mid \pi_z, s_{k,0}\right]\right\rangle + \mathbb{E}\left[\sum_{t=0}^{H-1} \gamma^t \Delta_r(s_{k,t}) \mid \pi_z, s_{k,0}\right] \qquad (92)$$

$$= \langle z_r, \psi^*(s_{k,0}, z)\rangle + \mathbb{E}\left[\sum_{t=0}^{H-1} \gamma^t \Delta_r(s_{k,t}) \mid \pi_z, s_{k,0}\right] \qquad (93)$$

$$\overset{A2'}{\leq} \langle z_r, \psi^*(s_{k,0}, z)\rangle + \frac{\zeta}{1-\gamma} \qquad (94)$$

We can then proceed along the main proof in Appendix A.4 as normal, substituting $\beta'$ for $\beta$ throughout. Given the suboptimal $\psi$ Eq. (43) now becomes

$$\langle \psi^\star(s_{k,0}, z_r), z_r\rangle \leq \text{UCB}_k(z_r) \leq \text{UCB}_k(z_k) = \langle \psi^\star(s_{k,0}, z_k), \tilde{z}_k\rangle. \qquad (95)$$

Then, in the bound on the instantaneous regret (Eq. (44)), we get an added term.

$$e_k = G_{k,z_r} - G_{k,z_k} \tag{96}$$

$$\leq \langle \psi^\star(s_{k,0}, z_r), z_r \rangle - \langle \psi^\star(s_{k,0}, z_k), z_r \rangle + \tfrac{2\zeta}{1-\gamma} \tag{97}$$

$$\leq \langle \psi^\star(s_{k,0}, z_k), \tilde{z}_k \rangle - \langle \psi^\star(s_{k,0}, z_k), z_r \rangle + \tfrac{2\zeta}{1-\gamma} \tag{98}$$

$$= \langle \psi^\star(s_{k,0}, z_k), \tilde{z}_k - z_r \rangle + \tfrac{2\zeta}{1-\gamma} \tag{99}$$

$$\overset{A2'}{=} \langle \psi(s_{k,0}, z_k), \tilde{z}_k - z_r \rangle + \langle \Delta_\psi(s_{k,0}, z_k), \tilde{z}_k - z_r \rangle + \tfrac{2\zeta}{1-\gamma} \tag{100}$$

$$\leq \langle \psi(s_{k,0}, z_k), \tilde{z}_k - z_r \rangle + \|\Delta_\psi(s_{k,0}, z_k)\|_{V_{k-1}^{-1}} \|\tilde{z}_k - z_r\|_{V_{k-1}} + \tfrac{2\zeta}{1-\gamma} \quad \text{Cauchy-Schwarz}$$

$$\leq \langle \psi(s_{k,0}, z_k), \tilde{z}_k - z_r \rangle + \tfrac{\xi}{\sqrt{\lambda}} \cdot 2\beta_k' + \tfrac{2\zeta}{1-\gamma}. \tag{101}$$

Where we used

$$\|x\|_{V_{k-1}^{-1}} = \sqrt{x^\top V_{k-1}^{-1} x} \leq \sqrt{\lambda_{\max}(V_{k-1}^{-1}) x^\top x} \leq \tfrac{1}{\sqrt{\lambda}} \|x\|_2 \tag{102}$$

in the last step.[8]

We can treat the added term $2\beta_k' \tfrac{\xi}{\sqrt{\lambda}} + \tfrac{2\zeta}{1-\gamma}$ separately throughout, bounding it by

$$2\beta_n' n \tfrac{\xi}{\sqrt{\lambda}} + \tfrac{n\zeta}{1-\gamma} \tag{103}$$

after summing over the episodes. As a result, inserting our new $\beta_n'$ in the regret bound, we obtain

$$R_n \leq 2\alpha \left( \sqrt{\lambda}S + \sigma \sqrt{d \log\left(\frac{d\lambda + nHL^2}{d\lambda}\right) + 2\log n} + \underbrace{\frac{\sqrt{2}L\zeta}{\sqrt{\lambda}} \sqrt{ndH \log\left(\frac{d\lambda + nHL^2}{d\lambda}\right)}}_{\text{new}} \right)$$

$$\times \left( \sqrt{2nd \log\left(\frac{d\lambda + n\frac{L^2(1-\gamma^H)^2}{(1-\gamma)^2}}{d\lambda}\right)} + \underbrace{\frac{2\xi n}{\sqrt{\lambda}}}_{\text{new}} \right)$$

$$+ \underbrace{\frac{n\zeta}{1-\gamma}}_{\text{new}} \tag{104}$$

Because of the new term appearing in $\beta'$ and the final constant term, the regret bound is no longer sublinear, but instead

$$R_n \leq \tilde{\mathcal{O}}(d\sqrt{n}) + \zeta\tilde{\mathcal{O}}(nd) + \xi\tilde{\mathcal{O}}(n\sqrt{dn}) \tag{105}$$

which is also the final regret when incorporating the event $\mathcal{E}^C$. This bound indicates that, while the confidence set may shrink initially, the two mismatches may introduce an irreducible uncertainty in the size of the respective missmatches that cannot be resolved. Crucially, the suboptimality incurred is directly connected to the degree of mismatch considered in assumptions (A1') and (A2').

## A.6   USEFUL PROPERTIES

### A.6.1   DETERMINANTS

We list a few useful properties that are used in the proofs above

**Proposition 6.** *Let $\phi_1, \ldots, \phi_n$ be a sequence of vectors with $\|\phi_t\| \leq L$ for all $t = 1, \ldots, n$, and $V_t = \lambda I + \sum_{s=1}^t \phi_s \phi_s^\top$.*

$$\log\left(\frac{\det V_n}{\lambda^d}\right) \leq d \log\left(\frac{d\lambda + nL^2}{d\lambda}\right) \tag{106}$$

---

[8]This approximation of $\lambda_{\max}(V_{k-1}^{-1}) \leq 1/\lambda$ is rather crude and could be improved by investigating how the eigenvalues of $V_k$ evolve over time.

*Proof.* First, note that

$$\det(\lambda I) = \lambda^d \text{ and } \operatorname{Tr}(\lambda I) = d\lambda \tag{107}$$

For $V_n$ we have, using the AM-GM inequality,

$$\det V_n = \prod_i \lambda_i \leq \left(\frac{1}{d} \operatorname{Tr} V_n\right)^d \leq \left(\frac{\operatorname{Tr} \lambda I + nL^2}{d}\right)^d \tag{108}$$

because $\|\phi\| \leq L$. Thus we have

$$\log\left(\frac{\det V_n}{\lambda^d}\right) \leq d\log\left(\frac{d\lambda + nL^2}{d\lambda}\right) \tag{109}$$

$\square$

### A.6.2 ELLIPTICAL POTENTIAL LEMMA

We here state a standard result, commonly referred to as the elliptical potential lemma, for completeness.

**Lemma 1.** *Assume $V_0 \succ 0$ be positive definite and let $\phi_1, \ldots, \phi_n$ be a sequence of vectors with $\|\phi_t\| \leq L$ for all $t = 1, \ldots, n$, and $V_t = V_0 + \sum_{s=1}^t \phi_s \phi_s^\top$.*

$$\sum_{t=1}^n \min\{1, \|\phi_i\|_{V_{t-1}^{-1}}^2\} \leq 2\log\left(\frac{\det V_n}{\det V_0}\right) \tag{110}$$

*Proof.* First, note that we have for $u \geq 0$

$$\min\{1, u\} \leq 2\log(1 + u) \tag{111}$$

and use it to get

$$\sum_{t=1}^n \min\{1, \|\phi_t\|_{V_{t-1}^{-1}}^2\} \leq 2\sum_{t=1}^n \log\left(1 + \|\phi_t\|_{V_{t-1}^{-1}}^2\right) \tag{112}$$

Then, notice that for $t \geq 1$ we have

$$V_t = V_{t-1} + \phi_t \phi_t^\top = V_{t-1}^{1/2}\left(I + V_{t-1}^{-1/2}\phi_t \phi_t^\top V_{t-1}^{-1/2}\right) V_{t-1}^{1/2} \tag{113}$$

so, taking the determinant on both sides we get

$$\det V_t = \det V_{t-1} \cdot \det\left(I + V_{t-1}^{-1/2}\phi_t \phi_t^\top V_{t-1}^{-1/2}\right) = \det V_{t-1}\left(1 + \|\phi_t\|_{V_{t-1}^{-1}}^2\right) \tag{114}$$

where we used $\det(A + B) = \det A \cdot \det B$ and $\det(I + uu^\top) = 1 + \|u\|^2$. Telescoping the previous result, we get

$$\det V_n = \det V_0 \cdot \prod_{t=1}^n \left(1 + \|\phi_t\|_{V_{t-1}^{-1}}^2\right) \tag{115}$$

which implies

$$\sum_{t=1}^n \log\left(1 + \|\phi_t\|_{V_{t-1}^{-1}}^2\right) = \log\left(\frac{\det V_n}{\det V_0}\right) \tag{116}$$

which proves the claim. $\square$

### A.6.3 UCB OPTIMISATION

We here state the derivation of a well-known property of LinUCB for completeness.

**Remark 1.** *We want to show that*

$$\arg\max_{z \in \mathcal{C}_t} \max_{w \in \mathcal{C}_t} w^\top \psi(s_t, z) = \arg\max_{z \in \mathcal{C}_t} \psi(s_t, z)^\top \hat{z}_{t-1} + \beta_t \|\psi(s_t, z)\|_{V_{t-1}^{-1}} \tag{117}$$

*Focusing on the inner* max *term, we have*

$$\max_{w \in \mathcal{C}_t} w^\top \psi(s_t, z) = \max_{\|w - \hat{z}_t\|_{V_{t-1}} \leq \beta_t} w^\top \psi(s_t, z) \tag{118}$$

$$= \hat{z}_t - 1^\top \psi(s_t, z) + \max_{\|u\|_{V_{t-1}} \leq \beta_t} u^\top \psi(s_t, z) \tag{119}$$

$$\leq \hat{z}_t - 1^\top \psi(s_t, z) + \max_{\|u\|_{V_{t-1}} \leq \beta_t} \|u\|_{V_{t-1}} \|\psi(s_t, z)\|_{V_{t-1}^{-1}} \tag{120}$$

$$= \hat{z}_t - 1^\top \psi(s_t, z) + \beta_t \|\psi(s_t, z)\|_{V_{t-1}^{-1}} \tag{121}$$

*where the equality is attained by*

$$u^\star = \beta_t \frac{V_{t-1}^{-1} \psi(s_t, z)}{\|\psi(s_t, z)\|_{V_{t-1}^{-1}}} \tag{122}$$

## B ADDITIONAL EXPERIMENTS

### B.1 HOW MUCH COMPUTE DOES OPTI-BFM NEED?

To estimate the compute cost of OpTI-BFM, we measure the time for the action selection and update steps on an Nvidia RTX 4090, skipping the first few calls to avoid measuring JIT compilation time. Table 1 shows that OpTI-BFM and OpTI-BFM-TS are about 5x and 4x slower than running just the policy (Oracle) respectively.

Table 1: Computational cost of OpTI-BFM compared to running just the policy on an Nvidia RTX 4090 GPU. OpTI-BFM is about 5x and OpTI-BFM-TS about 4x slower.

|  | Oracle ($\pi_{z_r}$) | OpTI-BFM | OpTI-BFM-TS |
|---|---|---|---|
| Time per Step | 0.772 ms | 3.567 ms | 2.756 ms |
| Frequency | 1386 Hz | 280 Hz | 363 Hz |

### B.2 REQUESTING REWARD LABELS EXPLICITLY

We have so far assumed that each environment interaction provides a reward label, and is thus synonymous with the labeling cost. One could also consider a setting where the agent can decide whether it wants to observe the reward label in each transition. A very simple approach in this setting is to threshold the D-gap (Kiefer & Wolfowitz, 1960; Lattimore & Szepesvári, 2020), which describes how much the confidence ellipsoid shrinks for a new $\phi_t$.

$$\Delta(\phi_t) = \log \det(V_{t-1} + \phi_t \phi_t^\top) - \log \det V_{t-1} = \log \left(1 + \|\phi_t\|_{V_{t-1}^{-1}}^2\right) \tag{123}$$

This approach can be directly applied to OpTI-BFM by introducing a hyper-parameter $\kappa$ and only requesting $r_t$ if $\Delta(\phi_t) \geq \kappa$. We evaluate labeling efficiency for this variant: for each labeling budget $n$ (# Samples), we measure average rewards at the time step $t$ in which the budget is exhausted, i.e. $\frac{1}{t} \sum_{i=1}^{t} r_t$. This is presented in Fig. 7 (see also Fig. 19 and Fig. 20 for per-task results). We observe that $\kappa$ trades-off environment interaction and labeling cost: for higher $\kappa$, good performance is achieved with fewer labels, but more episodes are required to request the same number of labels. Given a specific cost for labels, a slightly more involved thresholding scheme (Tucker et al., 2023) could potentially be used to decide the threshold apriori. Surprisingly, this variant of OpTI-BFM can maintain the performance of the main algorithm, especially in easier tasks, while reducing the amount of labeled data by one order of magnitude.

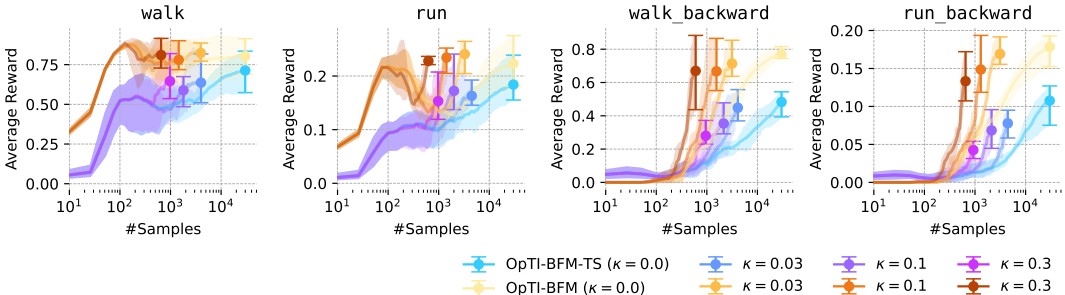

Figure 7: Average reward in Cheetah at different numbers of requested reward labels (# Samples) of our main methods for different information thresholds $\kappa$. We stop interaction after 30k environment steps. $\kappa$ trades-off interaction cost with labeling cost. More than one order of magnitude less reward labels can still result in the same performance in easier tasks! We report per method results of all environments in Fig. 20 and Fig. 19

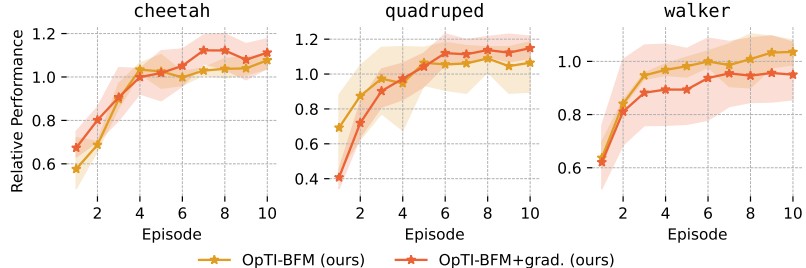

Figure 8: Relative performance after different # of environment interactions of OpTI-BFM and OpTI-BFM +grad; the latter is a variant that optimizes the USB objective through 8 gradient steps. Performances are similar; we report per-task results in Fig. 18 in Appendix C.

### B.3    FIRST-ORDER OPTIMIZATION OF THE UCB OBJECTIVE

This section introduces a variant of OpTI-BFM which uses gradient ascent to optimize the UCB objective:

$$\arg\max_{z \in \mathcal{Z}} \psi(s_t, z)^\top \hat{z}_{t-1} + \beta_t \left\| \psi(s_t, z) \right\|_{V_{t-1}^{-1}} , \qquad (124)$$

as opposed to the random shooting approach we describe in Section 4. We validate this variant empirically on `walker` tasks, relying on the Adam (Kingma & Ba, 2017) optimizer with a learning rate of $0.1$ to perform $2, 4, 8,$ or $16$ gradient steps in each step, and maintaining the optimizer state throughout. At each step, we utilize $\hat{z}_{t-1}$ to warm-start the optimization procedure. We report overall return curves in Fig. 8, and an ablation of compute costs in Fig. 9. We observe that, given a sufficient number of gradient steps, this variant approaches the main instantiation of our method in performance. However, compute requirements for gradient ascent scale linearly with the number of gradient steps, and surpass those of random shooting. Considering that state embeddings are, for most BFMs, relatively low-dimensional (e.g. $d = 50$), we conclude that random shooting approach should be preferred, as it is also less prone to fall into local optima. For very high-dimensional task spaces, first-order optimization remains however a promising option.

### B.4    OPTI-BFM FOR LOCO-NAVIGATION

To evaluate effectiveness beyond the locomotion experiments presented in the main text, we evaluate OpTI-BFM in `antmaze-medium-navigate-v0`, from the OGBench (Park et al., 2024a) benchmark. By default, this environment defines an indicator function reward that is 1 if the piloted quadruped reaches a certain goal position, at which point the episode terminates immediately. To make this environment consistent with our theory, we change the task to "holding" the goal position: the episode does not terminate until 1000 steps are accumulated. Furthermore, we slightly increase the radius of the sparse reward to cover the maze cell containing the goal, as seen in Fig. 10b in green, as reaching the goal exactly is otherwise challenging during online learning. As is common in this

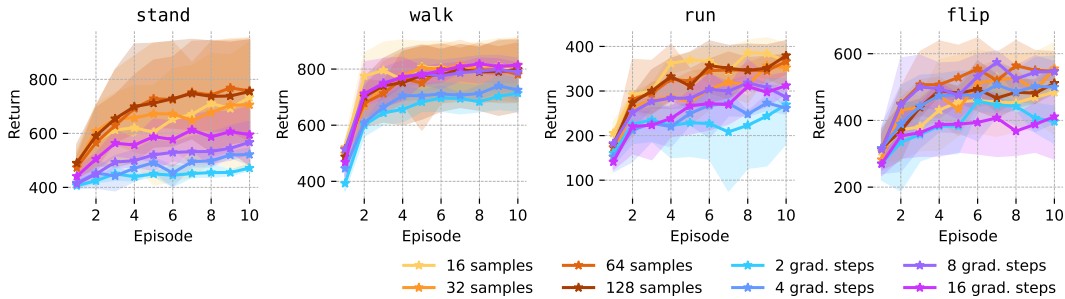

(a) Absolute performance for different # of samples or gradient steps.

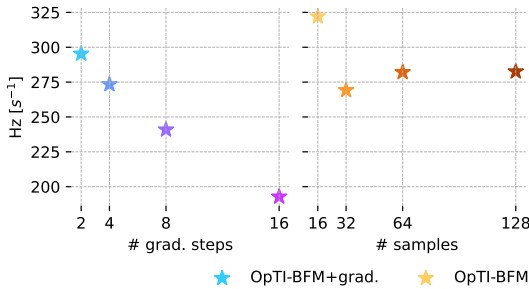

(b) Inference speed in Hz for different # of samples or gradient steps.

Figure 9: Performance and inference speed of OpTI-BFM, and a gradient-based variant (OpTI-BFM +grad). The performance gap is moderate, but the computational cost of the gradient-based variant is generally higher.

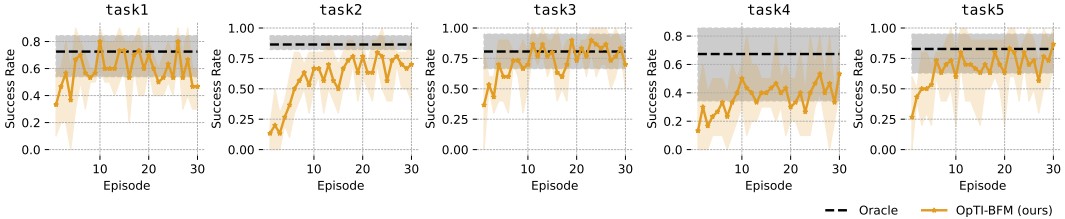

(a) Success Rate over 30 episodes.

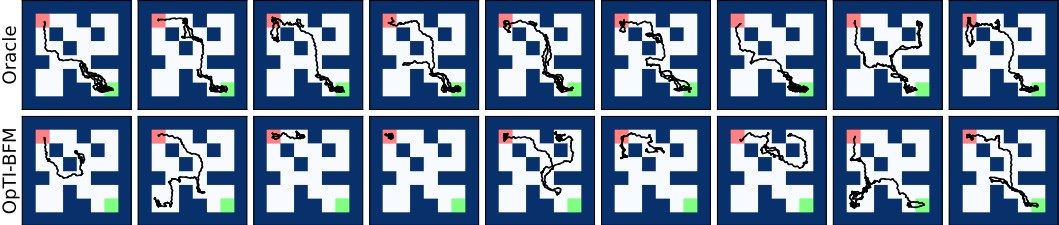

(b) The first 9 trajectories (left-to-right) of Oracle and OpTI-BFM in Task 2. The initial state is in the top left (red) and the goal is to reach and stay in the cell in the bottom right (green), which assigns a reward of 1 at each step. OpTI-BFM explores the whole maze before finally exploiting the reward it has seen in the green cell. We attribute any perceived exploration behavior of the Oracle to the high behavior cloning coefficient that is necessary to train policies that perform well in this setting.

Figure 10: OpTI-BFM can infer a goal-reaching task in the OGBench (Park et al., 2024a) loconavigation environent `antmaze-medium-navigate-v0`.

benchmark (Park et al., 2024a), we introduce a behavior cloning coefficient of 0.01 to the actor loss in pre-training, as the pre-training data contains near-expert navigation behavior as opposed to the high-coverage exploration data in ExORL. Results shown in Fig. 10 are consistent with our main evaluation: OpTI-BFM can reach near-Oracle performance within a handful of episodes.

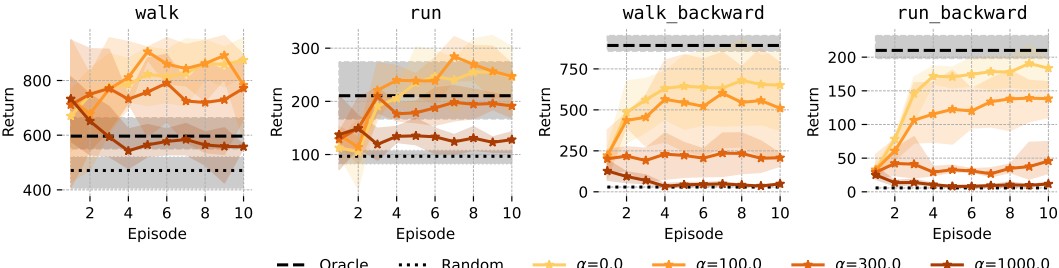

Figure 11: Performance of OpTI-BFM over 10 episodes in the Cheetah environment with a mismatch between $\psi$ and $\phi$ (see Eq. (125)) of magnitude $\alpha$. Performance deteriorates as $\alpha$ increases.

### B.5 OpTI-BFM with Inaccurate Successor Features

To investigate how robust OpTI-BFM is to violations of assuption A1, we evaluate its performance when introducing a systematic mismatch between the discounted sum of $\phi$ and $\psi$. Concretely, we let OpTI-BFM use

$$\psi'(s_t, z) = \psi(s_t, z) + \alpha \cdot \text{MLP}(z; \theta) \tag{125}$$

where MLP is a two layer network with ReLU activation and L2-normalized outputs, and $\theta$ are randomly sampled weights, which we resample at the start of task inference. This additional MLP introduces a systematic, $z$-dependent bias, whose magnitude may be controlled through the hyperparameter $\alpha$. To avoid confounding factors as much as possible, we adopt an optimization procedure that samples $z$ from the whole $\mathcal{Z}$ to optimize the UCB objective, increasing the number of samples to $n = 512$. As expected, Fig. 11 shows that performance deteriorates as we increase $\alpha$; very large values of $\alpha$ ($\approx 1000$) are necessary to render OpTI-BFM completely uninformative, and approach performance of the random baseline. As the norm of $\psi$ is generally bounded by $\frac{1}{1-\gamma}\sqrt{d} \approx 350$, these constitute significant perturbations.

### B.6 OpTI-BFM with Non-linear Rewards

This section investigates how a violation of assumption A2, i.e. linearity of rewards in the features $\phi$, impacts performance of OpTI-BFM. In order to isolate this effect as much as possible from other factors (e.g., how hard the task specified by a reward function is), we consider a family of reward functions with increasingly larger orthogonal components to the feature space. Starting from an existing reward function $r(s)$, we project it to the feature space, obtaining $z_r$, and then extract its orthogonal component $e(s) = r(s) - \phi(s)^\top z_r$ by training a small network. We can then control linearity of rewards and investigate performance under the reward function

$$r_\alpha(s) = z_r^\top \phi(s) + \alpha \cdot e(s). \tag{126}$$

Note that $r_0(s) = z_r^\top \phi(s)$ which has 0 projection error, and $r_1(s)$ recovers $r(s)$. While this technique can approximately disentangle reward components, for $\alpha > 1$ the reward function might increase in scale: episodic performance might thus actually grow with projection error. Furthermore, the function approximation of $e$ might be inaccurate, meaning that $\alpha$ does not allow directly tuning the projection error. We can nonetheless compare the return of OpTI-BFM (upon convergence, i.e. at its 10th episode) with that of the Oracle, over a range of mean absolute projection errors, as Fig. 12. While the results are increasingly noisy, as the error increases far beyond ranges encountered for existing rewards, OpTI-BFM seems to suffer slightly more in specific tasks—generally achieving worse performance compared to the Oracle for similarly misaligned reward functions. This may be explained by the fact that OpTI-BFM relies on linearity for both data collection and task inference, while the Oracle relies on independently collected data.

### B.7 Can OpTI-BFM learn from noisy rewards?

In this section we evaluate the performance of OpTI-BFM under noisy reward feedback. To this end, we add zero-mean Gaussian noise to the observed rewards with different standard deviations $\sigma$. As predicted by Eq. (66), we see in Fig. 13 that convergence is slower for higher standard deviations. Adjusting the hyper-parameter $\beta = 10$ for $\sigma = 10$ did improve performance slightly, but with a noise level that is 10x higher than the reward of the environment $r_t \in [-1, 1]$ convergence remains slow.

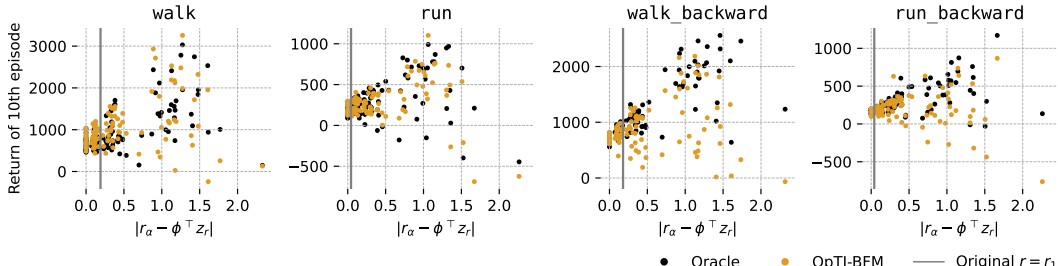

Figure 12: Return of the 10th episode of Oracle and OpTI-BFM in the Cheetah environment when artificially increasing or decreasing the projection error of the reward function onto $\phi$ through a learned network.

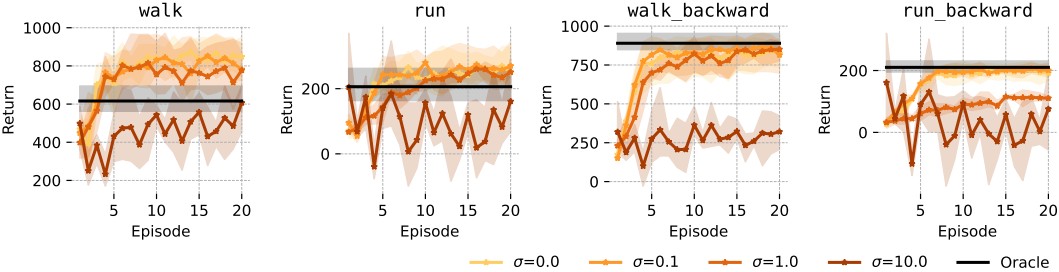

Figure 13: Performance of OpTI-BFM over 10 episodes in the Cheetah environment when adding Gaussian noise with different standard deviations to the rewards it observes. Convergence slows down as the noise is increased.

## C   FULL EXPERIMENTAL RESULTS

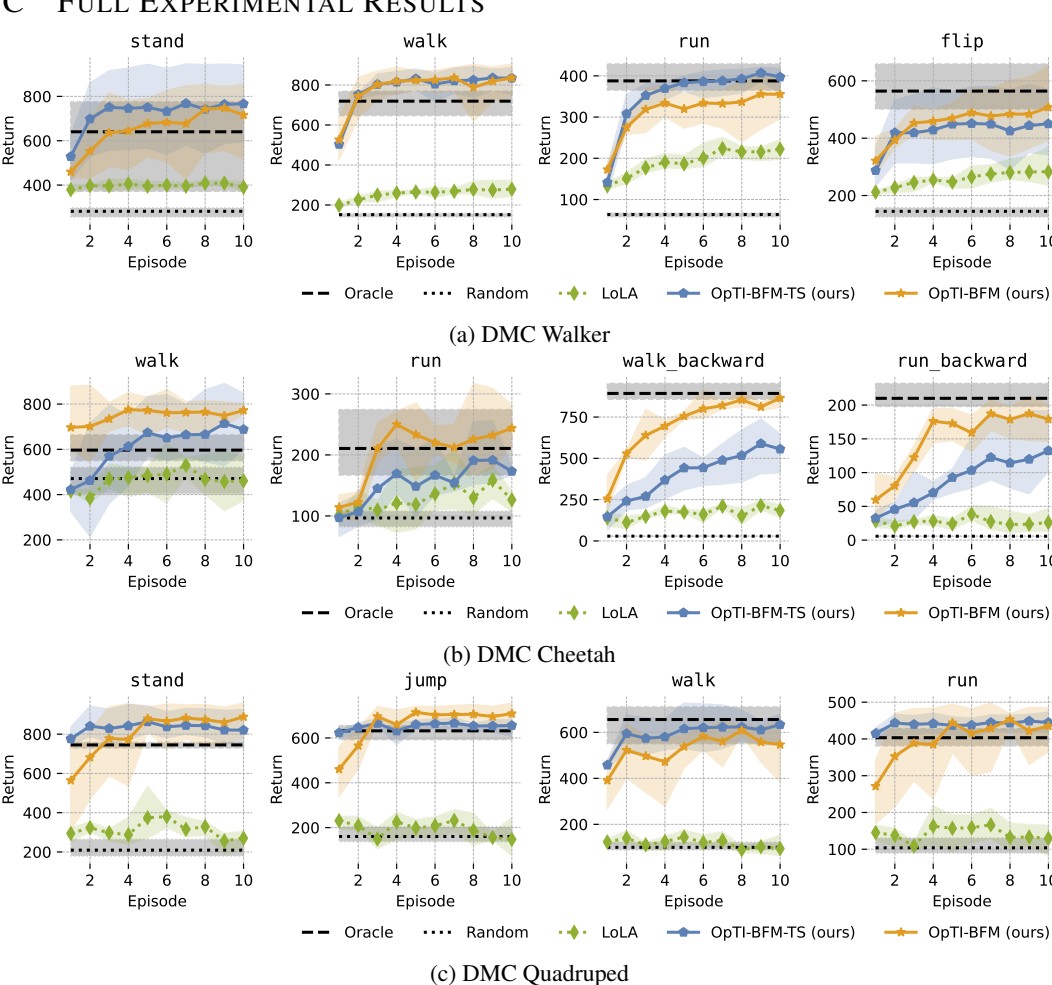

Figure 14: Episode Return over 10 episodes (10k steps) of interaction in DMC.

## D   TRAINING PROTOCOL

### D.1   FORWARD BACKWARD FRAMEWORK

We use the forward backward (FB) framework (Touati & Ollivier, 2021) as the base BFM throught our experiments. To fully explain FB, we quickly introduce the successor measure (Dayan, 1993; Blier et al., 2021) $M^\pi$; it is defined as

$$M^\pi(s_0, a_0, X) = \sum_{t \geq 0} \gamma^t \Pr[s_t \in X | s_0, a_0, \pi]. \tag{127}$$

and can be thought of as the discounted state occupancy of a policy $\pi$ when starting in $s_0, a_0$. FB then learns the low-rank decomposition of the successor measure density w.r.t. the empirical dataset measure $\mathcal{D}_{\text{train}}(ds)$:

$$M^{\pi_z}(s, a, ds^+) = F(s, a, z)^\top B(s^+) \mathcal{D}_{\text{train}}(ds^+). \tag{128}$$

where the policy should satisfy

$$\pi_z(a|s) > 0 \implies a \in \arg\max_a F(s, a, z)^\top z. \tag{129}$$

The successor measure and successor features are closely related, since we have that

$$\psi^\pi(s, a) = \int M^\pi(s, a, ds^+) \phi(s^+). \tag{130}$$

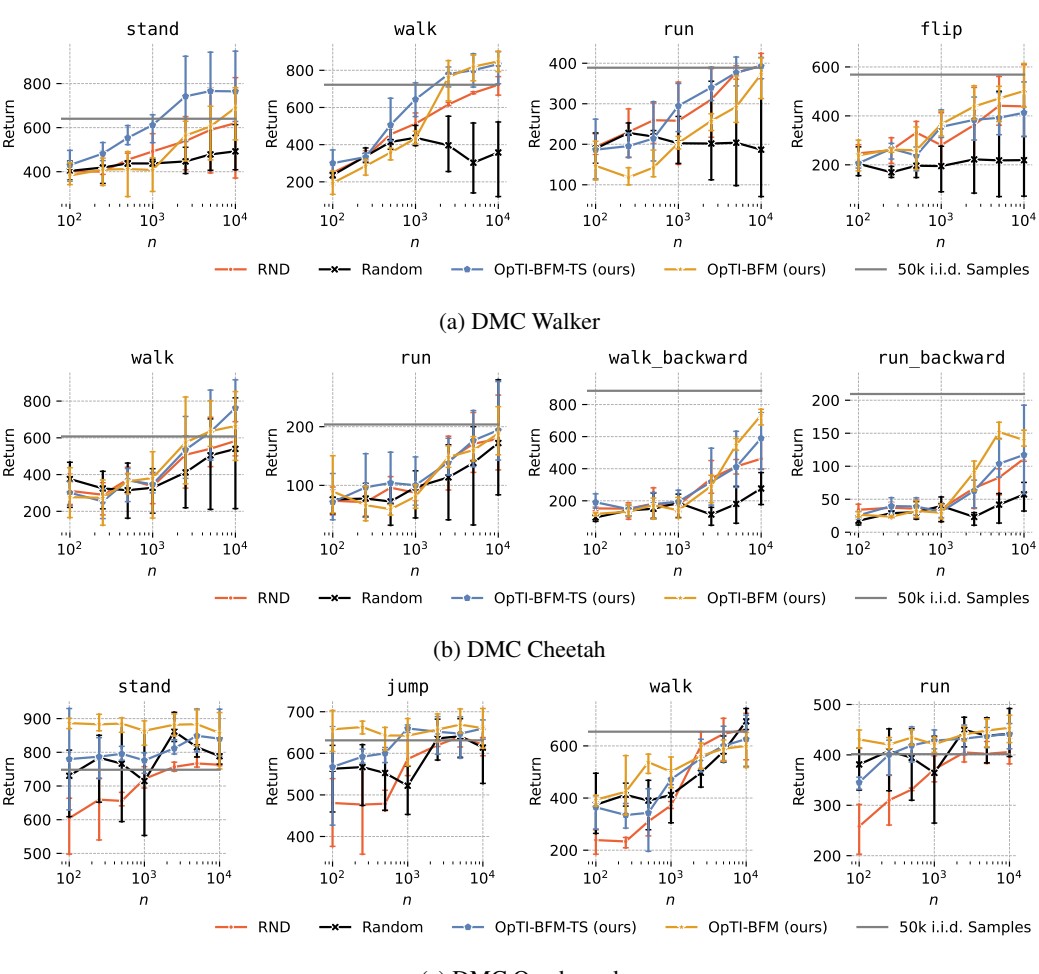

Figure 15: Per task results of Fig. 3. We show absolute performance here and include the Oracle performance (gray line).

Following Touati & Ollivier (2021) (Theorem 13), we thus have for FB

$$\int B(s^+)M_z^\pi(s,a,ds^+) = \int B(s^+)B(s^+)^\top \mathcal{D}_{\text{train}}(ds^+)F(s,a,z) \tag{131}$$

$$= \mathbb{E}_{s \sim \mathcal{D}_{\text{train}}}[B(s^+)B(s^+)^\top]F(s,a,z) \tag{132}$$

$$= (\text{Cov}_{\mathcal{D}_{\text{train}}}B)F(s,a,z) \tag{133}$$

so $F(s,a,z)$ are SFs of features $\phi(s) = (\text{Cov}_{\mathcal{D}_{\text{train}}}B)^{-1}B(s)$. Task inference for FB with labeled dataset $\mathcal{D}$ then effectively becomes

$$z_r = (\text{Cov}_{\mathcal{D}}\phi)^{-1}\mathbb{E}[\phi(s)r(s)] \tag{134}$$

$$= \left(\text{Cov}_{\mathcal{D}}\left((\text{Cov}_{\mathcal{D}_{\text{train}}}B)^{-1}B\right)\right)^{-1}(\text{Cov}_{\mathcal{D}_{\text{train}}}B)^{-1}\mathbb{E}[B(s)r(s)] \tag{135}$$

$$= (\text{Cov}_{\mathcal{D}_{\text{train}}}B)(\text{Cov}_{\mathcal{D}}B)^{-1}(\text{Cov}_{\mathcal{D}_{\text{train}}}B)(\text{Cov}_{\mathcal{D}_{\text{train}}}B)^{-1}\mathbb{E}[B(s)r(s)] \tag{136}$$

$$= (\text{Cov}_{\mathcal{D}_{\text{train}}}B)(\text{Cov}_{\mathcal{D}}B)^{-1}\mathbb{E}[B(s)r(s)] \tag{137}$$

which is consistent with Touati & Ollivier (2021) (Proposition 15). In practice we pre-compute $(\text{Cov}_{\mathcal{D}_{\text{train}}}B)$ with 50k samples from $\mathcal{D}_{\text{train}}$ after training.

## D.2 PRE-TRAINING

As is the standard in zero-shot RL benchmarks (Touati et al., 2023; Agarwal et al., 2025b), we train our FB model using an offline dataset collected with RND (Burda et al., 2018; Yarats et al., 2022) consisting of 10M transitions.

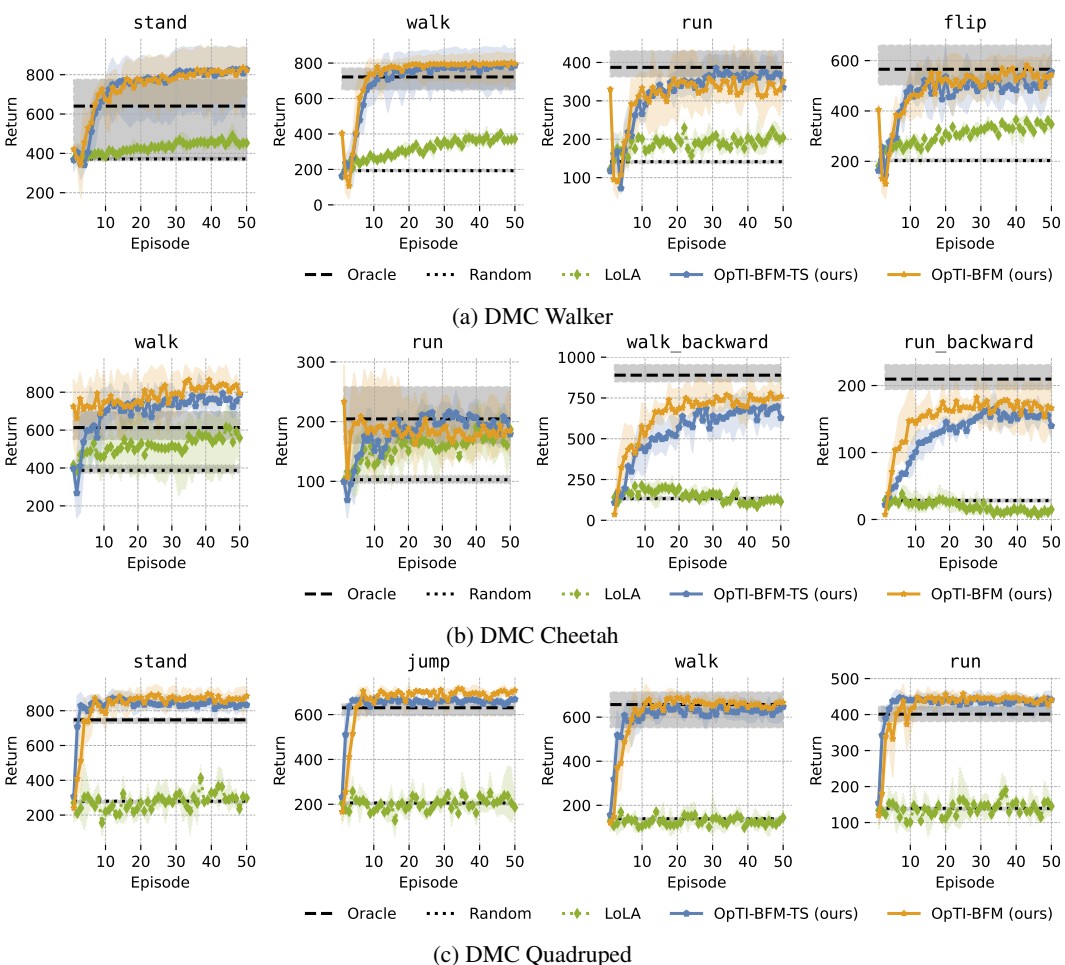

Figure 16: Episode Return over 50 episodes (50k steps) of interaction in DMC when only allowing switching of the task embedding at the start of episodes. Note that LoLA resamples its task embedding every 50-250 steps (hyper-parameter).

FB is trained using 3 losses. Firstly, the successor measure loss (Blier et al., 2021)

$$\mathcal{L}_{FB}(z) = \mathbb{E}_{\substack{s,a,s' \sim \mathcal{D}_{\text{train}} \\ a' \sim \pi_z(\cdot|s')}} \mathbb{E}_{s^+ \sim \mathcal{D}_{\text{train}}} \Bigg[ -2F(s,a,z)^\top B(s') \tag{138}$$

$$+ \left( F(s,a,z)^\top B(s^+) - \gamma \overline{F}(s',a',z)^\top \overline{B}(s^+) \right)^2 \Bigg], \tag{139}$$

where $\overline{F}, \overline{B}$ are target networks, secondly, an orthogonality regularizing loss on $B$

$$\mathcal{L}_{\text{ortho}} = \mathbb{E}_{\substack{s \sim \mathcal{D}_{\text{train}} \\ s' \sim \mathcal{D}_{\text{train}}}} \left[ -2 \|B(s)\|_2^2 + B(s)^\top B(s') \right], \tag{140}$$

and thirdly a DDPG-style loss for $\pi$:

$$\mathcal{L}_\pi(z) = \mathbb{E}_{s \sim \mathcal{D}_{\text{train}}} \mathbb{E}_{a \sim \pi_z(\cdot|s)} \left[ -F(s,a,z)^\top z \right], \tag{141}$$

using the reparameterization trick for Gaussian policies.

### D.3 ARCHITECTURE AND HYPER-PARAMETERS

We choose to follow the implementation details of the most recent work on FB (Tirinzoni et al., 2025), specifically their implementation for DMC in the released code-base. The $B$-network is a 3-layer MLP. The $F$ is an ensemble of size 2 with each two 2-layer MLPs to encode the arguments $(s,a)$ and $(s,z)$ that are then concatenated and fed into another 2-layer MLP. $\pi$ uses two 2-layer MLPs to

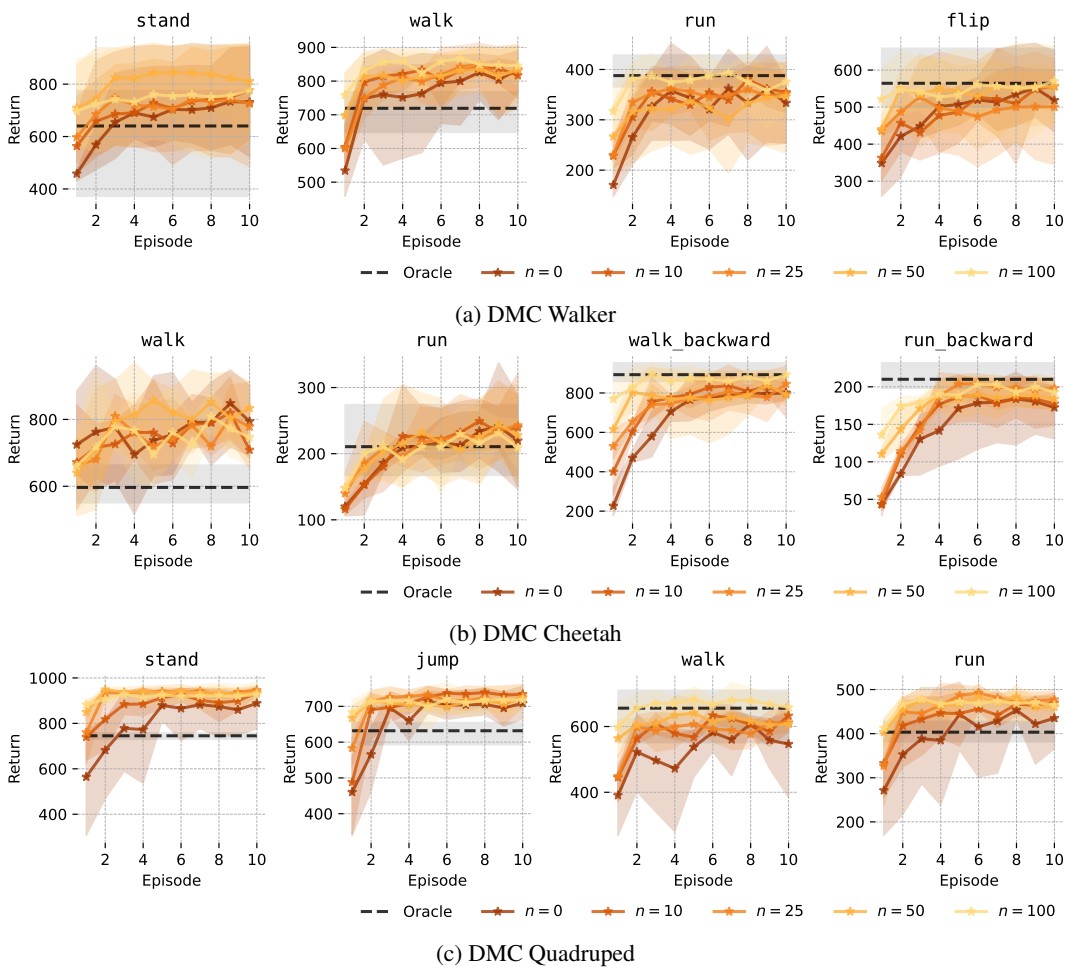

Figure 17: Return of OpTI-BFM when warm-starting with $n$ i.i.d. labeled states from the pre-training dataset.

encode the arguments $s$ and $(s, z)$ and then also another 2-layer MLP to map from the concatenated layers to a mean $\mu$. The policy is then a truncated Gaussian with fixed standard deviation of $\sigma = 0.2$ that truncates at $1.5\sigma$. The rest of the pre-training hyper-parameters are listed in Table 2 below. The task embedding space of FB is the $d$-dimensional hyper-sphere since the optimal policy $\pi_z$ is invariant to the scale of the reward function. Note that OpTI-BFM cannot make this simplification because it tries to estimate the hidden parameter of the reward function and not the latent that is plugged into the policy. We can recover the latter easily with an L2 normalization.

## E EXPERIMENT PROTOCOL

We evaluate our methods on a common zero-shot RL benchmark (Touati et al., 2023; Agarwal et al., 2025b). The benchmark consists of the Walker, Cheetah, and Quadruped environments from DMC (Tassa et al., 2018) with four tasks (reward functions) each. Note that each task has randomized initial states, so when evaluating single episode performance, e.g. Fig. 15, we report the mean over 20 episodes. And when evaluating task inference performance over multiple episode, e.g., Figs. 2 and 16, we report the mean over 10 trials. All error-bars and shaded regions denote min-max-intervals over three training seeds around mean performance.

### E.1 CUSTOM WALKER VELOCITY TASKS

The custom tasks we implement for DMC Walker Fig. 5 consist of only the velocity tracking components of the stand, walk, and run tasks. Further note that, by default, higher velocity is always allowed: a running policy will also perform rather well in standing. For this reason, we modify

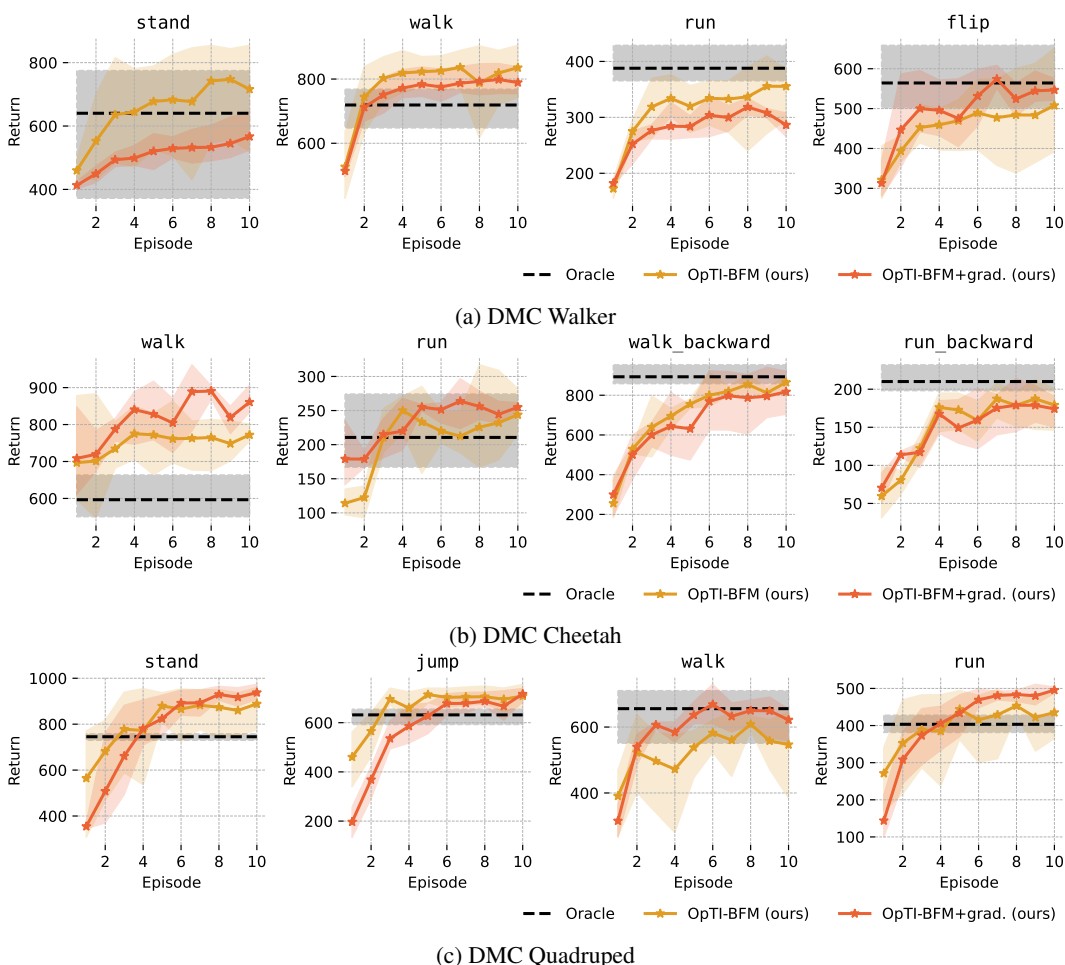

Figure 18: Return of OpTI-BFM and OpTI-BFM +grad. that use random-shooting and gradient ascent respectively to optimize the UCB objective. The two variants perform very similarly.

the velocity reward bonus so that it is 1 if the velocity matches the target exactly, and then linearly tapers to 0.5 when off by 2. From there, the reward then directly drops to 0.

## F INFERENCE HYPER-PARAMETERS

For each experiment and method we perform a grid-search over hyper-parameters. We then choose the hyper-parameters with highest overall cumulative return *per environment* for each method to report performance.

### F.1 OPTI-BFM

We found both OpTI-BFM and OpTI-BFM-TS to be very robust to the hyper-parameters we tested. For each method we consider a single hyper-parameter with three values each: For UCB, we test a fixed $\beta_t = \beta \in \{1.0, 0.1, 0.001\}$. For TS, we test $\sigma \in \{0.1, 0.001, 0.0001\}$. This means all other parameters where held constant. Specifically, $\rho = 1$ and $\lambda = 1$ if not reported otherwise (Fig. 5). The number of samples for the UCB optimization was $n = 128$ throughout if not specified otherwise.

### F.2 LOLA

For LoLA we consider a range of hyper-parameters to trade-off exploration, update frequency, update step-size, and variance in the gradient. We search all combinations of:

- horizon length and task embedding update rate $\{50, 100, 250\}$ as in (Sikchi et al., 2025);

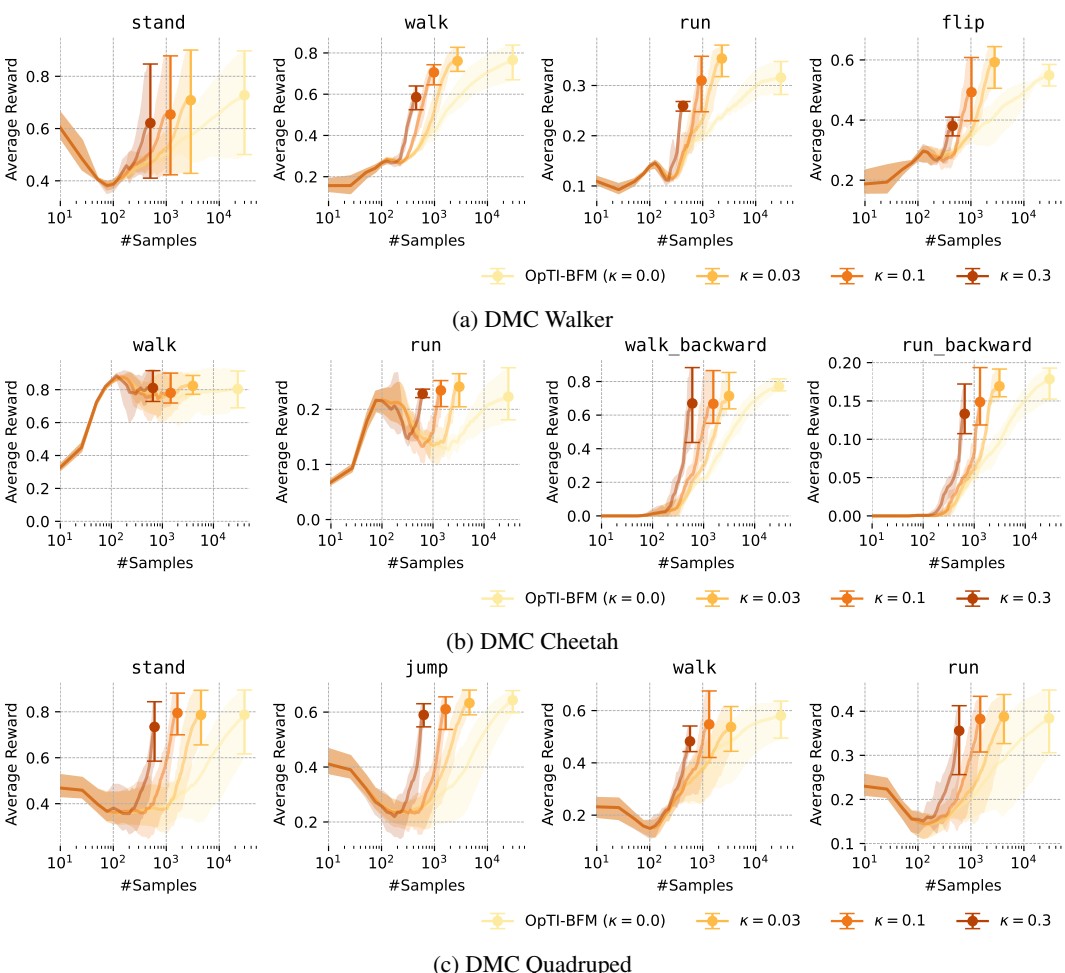

Figure 19: Average reward of OpTI-BFM at different numbers of requested reward labels (# Samples) for different information thresholds $\kappa$. We stop interaction after 30k environment steps. $\kappa$ trades-off interaction cost with labeling cost. More than one order of magnitude less reward labels can still result in the same performance in easier tasks.

- learning rate $\{0.1, 0.05\}$ as in (Sikchi et al., 2025);
- standard deviation of the Gaussian on the task embedding $\{0.05, 0.1, 0.2\}$.
- The batch size reported in Sikchi et al. (2025) is 5-10: we thus consider a replay buffer of 1000, which results in an effective batch size of 20, 10, and 4 for the respective update rates.

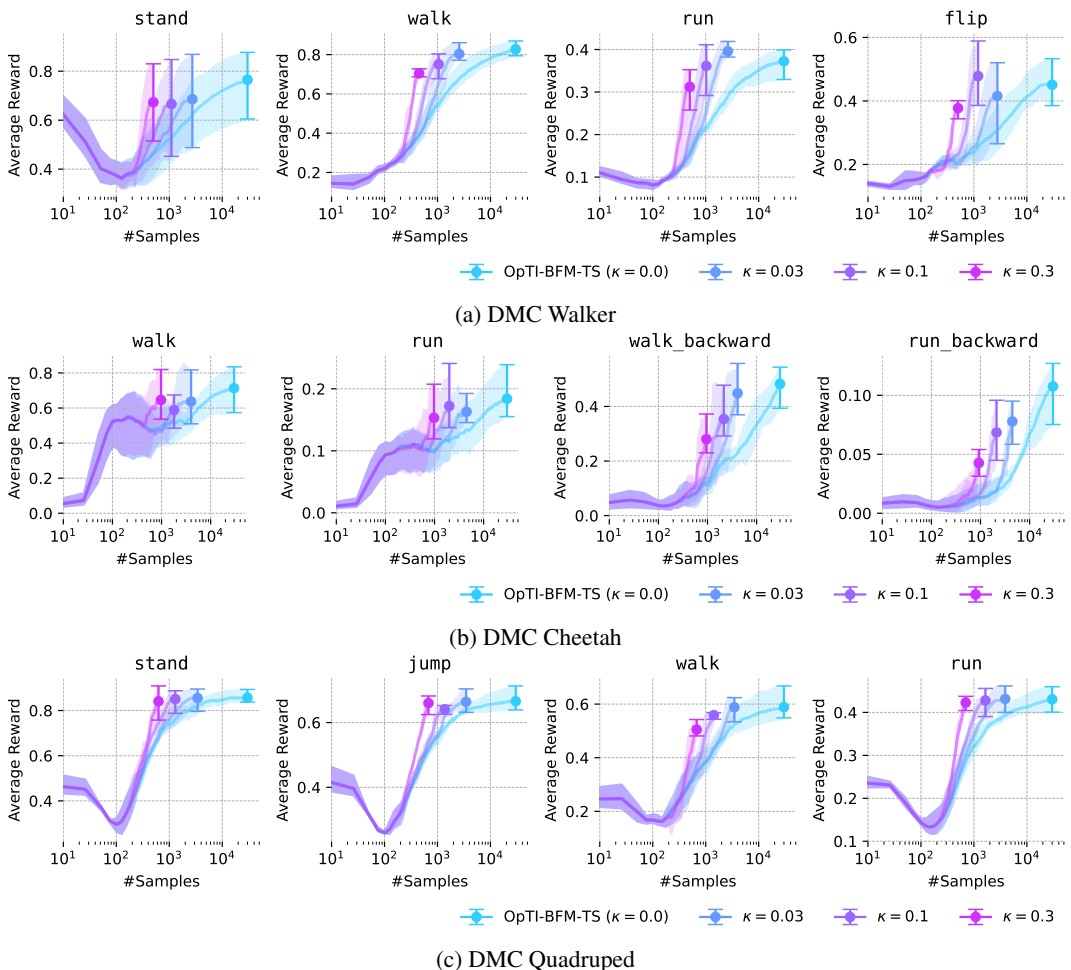

Figure 20: Average reward of OpTI-BFM-TS at different numbers of requested reward labels (# Samples) for different information thresholds $\kappa$.

Table 2: FB Hyper-Parameters.

| Name | Value |
|---|---|
| discount $\gamma$ | 0.98 |
| batch size | 1024 |
| # training steps | 2M |
| optimizer | adam |
| learning rate | 1e-4 |
| target network update factor | 0.01 |
| weight of $\mathcal{L}_{\text{ortho}}$ | 1.0 |
| $Q$-value penalty | 0.5 |
| fixed actor standard deviation | 0.2 |
| actor sample noise clipping | 0.3 |
| $z$ sampling | 50% $\mathcal{D}_{\text{train}}$ and 50% Random |
| dimension $d$ | 50 |
| $B$ network final activation | L2 normalization |
| $B$ network hidden dimension | 256 |
| $F$ network hidden dimension | 1024 |
| $\pi$ network hidden dimension | 1024 |
| all networks first activation | Layernorm + Tanh |
| all other activations | relu |

