# OpenReview forum: "Optimistic Task Inference for Behavior Foundation Models"
_ICLR.cc/2026/Conference — ICLR 2026 Oral_

### Official Review · Reviewer_pxZn · 2025-10-30

**Soundness:** 3
**Presentation:** 3
**Contribution:** 2
**Rating:** 6
**Confidence:** 4

**Summary:**

This paper proposes an ICRL (Interactive or In-Context Reinforcement Learning) algorithm based on Behavior Foundation Models (BFMs). The method performs online task inference by estimating the task embedding from observed rewards during inference, enabling dynamic policy adjustment. The authors provide partial theoretical analysis and show that the proposed method can achieve near-oracle performance within only a few episodes. Moreover, the approach can be extended to handle non-stationary reward settings.

**Strengths:**

1. The method is grounded on a solid theoretical foundation.

2. It is innovative and computationally efficient.

3. The approach is extensible and may inspire further research in this direction.

**Weaknesses:**

1. The experimental section only compares against Oracle and LoLA; it should include comparisons with other ICRL methods and evaluate optimization speed under out-of-distribution (OOD) settings.

2. In Appendix A5.3, line 1036, equation (84), the variable _x_ should likely be _ψ_.

3. In Algorithm 1, the formula for updating the estimator should reference the corresponding equation number for clarity.

**Questions:**

1. How does the proposed estimation procedure perform in sparse-reward settings (e.g., when the reward is only given at the end of an episode)?

---

> ### Author Response · Authors · 2025-11-20
> **Response to pxZn**
>
> Thank you for providing positive feedback and posing clear questions. We will address each comment individually.
>
> ### W1: Comparison to ICRL
>
> We find that applications of in-context reinforcement learning in the realm of BFMs has so far been limited: our main baseline (LoLa [1]) does not provide comparisons to established methods, and outperforms the baselines it in turn considers. While task inference is also implicitly performed by popular in-context RL methods (such as decision transformers [2]), their application in these settings is not straightforward: DTs traditionally rely on explicit reward labels and high-quality data, while BFMs are largely trained on reward-free, low-quality datasets, such as ExoRL [3]. For these reasons we are unsure about what would constitute a fair ICRL baseline in our setting; we are happy to follow any pointers in this regard.
>
> ### W2: Typo
> Thank you for pointing this out, we fixed both this $x$ and an indexing error in this equation.
>
> ### W3: Algorithm clarifications
> Following this suggestion, we have added an explicit update formula in the revision.
>
> ### Q1: Sparse rewards
> Opti-BFM can be generally applied with sparse rewards, both in time and over the state space. When rewards are sparse in time (i.e., feedback is only provided at the end of each episode), the variant of OpTI-BFM with episodic task-latent updates is directly applicable, as it resamples task embeddings only at the start of each episode. Incidentally, this is the variant for which our theoretical guarantees hold exactly. When rewards are provided at each step, but they are almost always zero, the main variant of OpTI-BFM can be directly applied. We demonstrate this in a new evaluation on antmaze in Appendix B.4, and confirm that our method remains capable of recovering oracle performance in a handful of episodes. As this environment is easy to interpret, we have included a visualization of the algorithm’s behavior as well.
>
> Thank you again for providing clear and actionable comments; we hope we were able to address all concerns, and remain available for further discussion.
>
> References:
> - [1] Sikchi et al., Fast Adaptation with Behavioral Foundation Models, RLC 2025
> - [2] Chen et al., Decision Transformer: Reinforcement Learning via Sequence Modeling, NeurIPS 2021
> - [3] Yarats et al.,  Don’t Change the Algorithm, Change the Data: Exploratory Data for Offline Reinforcement Learning, ICLR GPL 2022

---

> > ### Author Response · Authors · 2025-11-27
> >
> > Dear Reviewer,
> >
> > as we appreciate your valuable feedback, we would like to double-check whether our response was able to address your comments. In case anything remains unclear, or you would like to see additional experiments, please let us know. We are happy to answer any questions and further update the revision.

---

### Official Review · Reviewer_REKP · 2025-10-30

**Soundness:** 3
**Presentation:** 3
**Contribution:** 3
**Rating:** 6
**Confidence:** 4

**Summary:**

The paper adresses task inference for BFMs without relying on a labeled “task‑inference” dataset at test time. Instead of estimating a point task embedding offline, the agent actively interacts with the environment and updates a belief over reward parameters. The core idea is to view policy search over task embeddings as linear bandit optimization that comes from the insight that for USF-based BFMs, the relationship between successor features and returns is approximately linear. So the paper proposes OpTI‑BFM, which maintains a least‑squares estimate over the unknown reward weights, then selects the task embedding optimistically. It provides theoretical sublinear regret bounds $O(d\sqrt(n))$ demonstrates empirically that the method can identify tasks within 5 episodes on DMC benchmarks.

**Strengths:**

- One of the core original ideas is the new task‑space bandit formulation for BFMs. The paper formulates online task inference as linear bandit optimization in the task-embedding space: with well-trained USFs, the expected episode return is approximately linear in the successor features of the policy conditioned on a task embedding, i.e., $\mathbb{E}[\hat{G}_k \mid s_0, \pi_z] \approx \langle \psi(s_0, z), z_r \rangle$. This lets the agent choose $z$ using a rule over confidence sets on the unknown reward weights $z_r$ (Eqs.~(7)--(10), \S3.2). The ``two-context'' twist (estimating with features $\phi$ but acting with successor features $\psi$) is new in the BFM literature and differentiates the method from standard LinUCB.
- Another contribution shows that running least-squares on reward-level pairs $(\phi, r)$ yields tighter confidence sets than regressing on episode-level returns and empirical SFs $(\tilde{\psi}, \hat{G})$.
- The paper has efficient implementation details that are non-trivial. For example: the optimizer is computationally light and table 1 shows that the machinery is deployable in real-time control.
- The paper is mathematically sound and strong.

**Weaknesses:**

- The theory assumes perfect USFs and that the policy conditioned on $z$ is (near) optimal for reward $z$ (A1), strictly linear rewards with sub-Gaussian noise (A2), and an optimization oracle for Eq.~(10) (A3) and are introduced before Algorithm1 on p.5. In practice, USFs are learned with function approximation and the acquisition is solved by random shooting. The paper notes ``we found OpTI-BFM to perform well even when [A1--A2] are violated'' but does not quantify robustness to misspecification.
- Also the regret bounds are proven for a variant that only updates $z$ at episode starts (\S3.3), while the recommended/practical algorithm updates $z$ every step, which is empirically much better (Fig.4). So the bound doeen't exactly cover the method actually used. (p.5; Fig.4 p.8).

**Questions:**

- Add controlled experiments that systematically break assumptions. For example inject bounded bias into $\psi$ and report regret vs.\ SF error or create rewards with a tunable component orthogonal to $\phi$ or ablate sub-Gaussian noise level. Reporting regret/performance vs.\ the projection error $\|r - \phi^\top z\|$ would make the empirical section align with A2/A1.

- Provide either 1) a theoretical extension to per-step updates, or 2) a head-to-head comparison that keeps the episodic-only variant as the default and shows the practical gap in other settings too, with a candid discussion of why the bound should still be true.

---

> ### Author Response · Authors · 2025-11-20
> **Response to REKP**
>
> We would like to thank the reviewer for their detailed and clear comments. We believe that the points you raised are very interesting, and we are happy to provide a thorough discussion.
>
> ### W1&Q1: Assumptions A1&A2
>
> As suggested, we have added an analysis of performance degradation when key assumptions are violated as Appendix B.5, B.6, and B.7. We refer to the general response and to the revision for a complete discussion of these results. Additionally, we have studied formal consequences of violations of these assumptions, which we summarize in the general response, and describe at length in Appendix A.5.
>
> ### W2&Q2: Gap between theory and practice
>
> Our theoretical guarantees cover a variant of the algorithms that updates its task embedding on a per-episode basis. While this variant performs rather well (Figure 4), we find that per-step updates further accelerate convergence. Providing guarantees for the practical variant is however challenging unless strong assumptions are introduced, to the best of our knowledge. As we believe that this is an important comment, we provide more detailed comments in the general response. Empirically, if we understand correctly, a “head-to-head” comparison is already present as Figure 4. We apologize in case this is a misunderstanding, and would kindly ask for clarifications.
>
> Thank you again for your informed and precise review of our work. We are happy to receive further feedback on our responses, and to discuss any open questions.

---

> > ### Author Response · Authors · 2025-11-27
> >
> > Dear Reviewer,
> >
> > as we appreciate your valuable feedback, we would like to double-check whether our response was able to address your comments. In case anything remains unclear, or you would like to see additional experiments, please let us know. We are happy to answer any questions and further update the revision.

---

### Official Review · Reviewer_eb95 · 2025-11-01

**Soundness:** 3
**Presentation:** 3
**Contribution:** 4
**Rating:** 8
**Confidence:** 2

**Summary:**

The authors tackle a key limitation of behavioural foundation models (BFMs) based on Universal Successor Functions (USFs): the need for a dataset of labeled (state, reward) pairs. For cases where such data may be impractical or prohibitively costly to acquire, an alternative framework based on actively collecting a smaller amount of data online during deployment is explored.

To navigate this new framework the authors propose OpTI-BFM(Optimistic Task Inference for Behavioural Foundation Models) which leverages the linear relationship between features and rewards to update its belief over the space of rewards (i.e., incrementally improves its estimate of the task embedding using observed rewards). OpTI-BFM maintains a confidence ellipsoid over possible task embeddings and selects actions optimistically to efficiently explore the reward space.

Leveraging the fact that policy search for well-trained USF-based BFMs reduced to online optimisation of a linear function, a regret bound for OpTI-BFM in an episodic setting is established - the expected regret over n episodes $R_n\leq \mathcal{O}(d\sqrt{n})$. OpTI-BFM is evaluated empirically using a common zero-shot RL benchmark (ExORL) consisting of Walker, Cheetah and Quadruped environments with four reward functions each. OpTI-BFM achieves oracle performance (upper bound) within five episodes (5k steps) on all tasks, outperforming LoLA and the “Random” (lower bound) baselines.

**Strengths:**

The authors introduce a new framework for task inference in BFMs without labeled offline (state, rewards) data. In this framework, the relationship between BFM policy search and linear bandits is exploited to develop, and prove a regret bound for, the OpTI-BFM algorithm for online task inference. OpTI-BFM is timely and tackles the problematic requirement for labeled data with implications for many real world applications. The empirical results support the efficacy of OpTI-BFM in three standard zero-shot tasks (Walker, Cheetah, Quadruped), outperforming LoLA and reaching Oracle-level performance within 5 episodes.

**Weaknesses:**

- Whilst the experiment section is quite strong already it could be improved further if the authors are able to show the performance of OpTI-BFM on an alternative environment to those of the DeepMind Control suite. e.g. an alternate task with pixel observations.

- There is not much discussion of how OpTI-BFM could be deployed for real-world use. The authors say that their method would enable BFMs to work “beyond domains in which rewards are readily available”, but it is not obvious to me how it would interact with a real environment to get immediate reward labels. It would be appreciated if this could be explained further by the authors.

**Questions:**

Could the authors comment on whether OpTI-BFM generalises beyond continuous control environments?

---

> ### Author Response · Authors · 2025-11-20
> **Response to eb95**
>
> We would like to thank the reviewer for providing comprehensive feedback, and for the actionable suggestions, to which we will now respond.
>
> ### W1&Q1: Other Environments
> Following your suggestion, we have added an evaluation of OpTI-BFM on antmaze-medium, which focuses on navigation rather than simple locomotion. These results, which we report in Appendix B.4, largely align with those previously reported: OpTI-BFM reaches Oracle performance in a handful of trajectories.
> Extension to pixel-based tasks is a very interesting direction. At the time this work was submitted, applications of BFMs to visual domains had however been rather limited [1, 2]: reported oracle performance is rather low, leaving little signal for evaluating new algorithms.
> For this reason, while OpTI-BFM can be applied independently from the type of input,  we did not consider these environments for our empirical evaluation. Concurrent work [3] has however suggested that the forward-backward (FB) framework we use to train the BFM in our experiments shows strong performance in pixel-based environments, which makes it a promising application upon code release.
>
> ### W2: Real-world Deployment
> As a didactic example, we can consider the task of manipulation from pixels. In this case, a reward cannot be easily computed from states alone: the standard task inference for BFMs would require querying a reward model (either a foundation model, or an annotator on a relatively large dataset. Opti-BFM would reduce the number of reward labels required, and thus practical costs. When a foundation model is used for providing rewards, it may be queried on-the-fly. When a human annotator is available, and step-level feedback is less practical, they could directly label entire trajectories in-between rollouts (recovering the episode-level update variant of OpTI-BFM). Finally, relabeling costs may be further reduced by actively selecting the transitions to label, as shown in Appendix B.2.
> We hope this answers your questions and concerns. We remain available for follow-up discussions for the remainder of the rebuttal period.
>
> References:
> - [1] Park et al., Foundation Policies with Hilbert Representations, ICML 2024
> - [2] Jajoo et al., Regularized Latent Dynamics Prediction is a Strong Baseline For Behavioral Foundation Models, RLC IDGA 2025
> - [3] TD-JEPA: Latent-predictive Representations for Zero-Shot Reinforcement Learning; Bagatella et. al. ‘25

---

> > ### Author Response · Authors · 2025-11-27
> >
> > Dear Reviewer,
> >
> > as we appreciate your valuable feedback, we would like to double-check whether our response was able to address your comments. In case anything remains unclear, or you would like to see additional experiments, please let us know. We are happy to answer any questions and further update the revision.

---

### Official Review · Reviewer_afmy · 2025-11-01

**Soundness:** 3
**Presentation:** 4
**Contribution:** 3
**Rating:** 6
**Confidence:** 4

**Summary:**

This paper proposes OpTI-BFM, an optimistic decision criterion that directly models uncertainty over reward functions and guides BFMs in data collection for task inference. Authors frame this online task inference problem as a linear bandit problem and maintain a probabilistic belief (a confidence ellipsoid) over the true task embedding z_r by performing real-time least-squares regression on reward-level feedback. Experiments on zero-shot benchmarks show that OpTI-BFM matches or surpasses offline reward inference methods with much less data.

**Strengths:**

- Tackles a significant and practical bottleneck. Online task inference with only a few active-interaction episodes is important for real-world applications.

- Solid theoretical guarantees via connections to linear bandit algorithms.

- Good empirical performance with additional experiments and analysis, e.g. episode-level updates and non-stationary rewards.

**Weaknesses:**

- The paper operates under assumptions of a perfect BFM / successor feature model. It is unclear what would happen for the theoretical guarantees or how the empirical results would change with approximation errors.

- The formal regret bound is proven for an episodic-update variant of OpTI-BFM. However, the experiments (Sec.5.3, Fig. 4) show that the step-update variant is empirically superior and converges much faster. While it is a positive result that the practical algorithm is even better, it means the theory doesn't formally cover the best-performing algorithm presented.

- The optimistic search for z is currently done by random shooting and may fail for larger, more complicated spaces.

**Questions:**

- How is the method's robustness to misspecification? For example, how does OpTI-BFM perform if the true reward function r(s) has a significant non-linear component, or if the pre-trained BFM is of lower quality (violating A1)? How does its degradation compare to the offline "Oracle" regression, which would also suffer from this misspecification?

- How does the random shooting for UCB optimization scale? Have you explored the sensitivity to this number of samples? Would a gradient-based approach be more robust or scalable?

- Could the theoretical guarantees be extended to the step-level update case?

---

> ### Author Response · Authors · 2025-11-20
> **Response to afmy**
>
> Thank you for your thorough assessment of our submission, and for the interesting questions. We have clustered weaknesses and questions, which we will now address.
>
> ### W1&Q1: Assumptions
>
> Thank you for raising this important point. We would like to refer to the general response for a thorough discussion and empirical validation on robustness to misspecification.
> More in general, violations of Assumption A1 (i.e. imperfect successor features) would also impact the performance of the Oracle as far as policy extraction during training is concerned; however, the standard offline task inference procedure does not leverage successor feature estimates, and is thus robust to these inaccuracies. Violations of Assumption A2 (i.e. non-linearity of rewards) would instead degrade both the Oracle’s and OpTI-BFM’s performance, as studied in the newly introduced Appendix B.6. Artificially large violations will impact the latter in particular, as it also relies on A2 for data collection. We are happy to discuss these comments in further detail.
>
> ### W2&Q3: Step-level guarantees
>
> While guarantees cover episode-level updates, they do not extend to the (empirically better) variant employing step-level updates. Providing guarantees to the per-step update case remains challenging, to the best of our knowledge. We would like to again refer to the general response for a discussion of possible approaches and challenges, and we are open to discussing what type of techniques could ease this analysis.
>
> ### W3&Q2: Scaling of random shooting
>
> Similarly to recent works [1], we find that random shooting performs reasonably well when optimizing over task embeddings with a practical sampling budget. As suggested, we have added a sensitivity analysis to the number of samples in Appendix B.3, which suggests that OpTI-BFM continues to perform reasonably well at lower budgets. While sampling-based optimization might not scale well to very large latent spaces, BFMs usually operate over relatively compact feature spaces (e.g. BFM-Zero [2] is capable of full-body humanoid control, and relies on 256 dimensions); therefore, we believe that random shooting will remain a viable option. In any case, one may also employ a gradient-based alternative, which will potentially scale better with the dimensionality. We show this in newly added experiments in Appendix B.3: while performance of the method is similar, we find that the computational cost of gradient based techniques is slightly higher in these settings.
>
> We hope this fully addresses your questions and comments, and remain happy to engage in further discussion for the rest of the rebuttal period.
>
> References:
> - [1] Farebrother et al., Temporal Difference Flows, ICLR 2025
> - [2] Li et al., BFM-Zero: A Promptable Behavioral Foundation Model for Humanoid Control Using Unsupervised Reinforcement Learning, arXiv 2025

---

> > ### Author Response · Authors · 2025-11-27
> >
> > Dear Reviewer,
> >
> > as we appreciate your valuable feedback, we would like to double-check whether our response was able to address your comments. In case anything remains unclear, or you would like to see additional experiments, please let us know. We are happy to answer any questions and further update the revision.

---

### Author Response · Authors · 2025-11-20
**General Response**

We would like to thank all reviewers for providing a positive and detailed evaluation of our submission. We are happy that our contribution is deemed “important”, “timely” or “innovative” (afmy, eb95, pxZn), supported by “solid theoretical guarantees” (afmy, eb95, REKP, pxZn) and  “good empirical performance” (afmy, eb95). We would like to use this general response to address two specific comments, and summarize the changes in the revision.

### Lifting assumptions

Our method operates under the assumptions that successor features are well-trained (A1), and the reward is well-captured by the features (A2). We are happy to comment on what happens when either of these two assumptions does not hold.

As reviewers afmy and REKP point out, and we mention on line 239, the assumption that USF training was perfect (A1) will almost certainly be violated when using neural networks.
Empirically, we find that OpTI-BFM is robust to USF inaccuracies, at least to the degree that arises in standard benchmarks (i.e., DMC). We further investigate the degree of robustness of OpTI-BFM in controlled experiments in a new ablation in Appendix B.5, and find that very large perturbations are necessary to render the method uninformative.
Formally, inaccuracies in the successor features results in an additional term in instantaneous regret (Equations 44, 98) in the worst case. We have added a formal study in Appendix A.5.

Assumption A2, which posits that the reward function is exactly linear in the features $\phi$, also does not hold in practice. Constructing a reward function with a strong orthogonal component to the feature basis is non-trivial: we have resorted to learning and scaling orthogonal components, producing the empirical evaluation in Appendix B.6, which studies performance degradation as the projection error increases. To complement this evaluation, we have also included a proxy empirical study, which, as suggested, injects noise to reward feedback (see Appendix B.7). Accordingly, we find that significant noise injection slows down convergence.
Formally, misalignment between feature and rewards breaks the elliptical potential lemma (Theorem 1), and renders analysis more challenging. However, we may extend results from the literature on misspecified linear bandits: misspecification results in regret terms, whose growth can be related to the degree of misspecification. We provide a detailed analysis in Appendix A.5.

### Per-step updates

Extending guarantees to the (empirically stronger) step-level variant of OpTI-BFM is an exciting prospect, but unfortunately this introduces a few additional challenges. The current proof, for instance, relies on episodic resets to an initial state that is not influenced by the policy in order to invoke results from contextual linear bandit theory. To the best of our knowledge, step-level guarantees would likely need to be derived from existing results in reinforcement learning settings, and thus deal with a much more complex analysis.
One way to obtain such guarantees would be to introduce strong smoothness assumptions: intuitively, if the posterior evolves slowly, and the agent’s behavior is smooth with respect to its task embedding, its trajectories would not significantly vary from those induced by episode-level updates. While this may produce a regret guarantee, proving better performance with respect to the episode-level counterpart would remain challenging. Intuitively, faster policy updates can induce lower regret, as the agent may react faster to new information, but guaranteeing that the feedback is “as useful” is the main challenge.
In short, we agree that this missing guarantee represents an important direction for future work, and we have highlighted this as such at the end of the main text on line 478. We are happy to consider and discuss any suggestions that might simplify this analysis.

### Changes

We would like to finally summarize the main topics of discussion, and corresponding updates to the revision (which we highlight in color).
- We extended the range of considered environments to consider a sparse-reward navigation problem in Appendix B.4.
- A new variant of OpTI-BFM relying on gradient-based optimization of the UCB objective is introduced and studied in Appendix B.3.
- We investigated the method’s behavior under violation of the two main assumptions (accuracy of successor features and linearity of rewards), both formally (Appendix A.5), and in controlled ablations (Appendix B.5, B.6 and B.7).
- We updated several parts of the main paper and appendix, following the reviewers’ suggestions.

We believe that these updates were instrumental in strengthening this work, and we would like to thank all reviewers for suggesting these extensions. We look forward to the upcoming discussion phase.

---

### Meta-Review · Area_Chair_XzZ2 · 2026-01-07

**Summary:**

This submission addresses a critical limitation in deploying Behavior Foundation Models (BFMs) based on Universal Successor Features (USFs): the standard requirement for a labeled dataset to perform offline task inference. The authors propose OpTI-BFM, a novel online task inference framework that formulates the search for the correct task embedding as a linear bandit problem. By leveraging the linear relationship between successor features and rewards, the method constructs confidence ellipsoids over task embeddings to guide exploration optimistically. The authors provide a rigorous theoretical foundation, establishing a regret bound for the episodic variant of the algorithm. Empirically, the method demonstrates impressive data efficiency on the ExORL benchmark (DMC Suite), recovering oracle-level performance within a handful of episodes and significantly outperforming baselines like LoLA and random search.

All reviewers gave positive initial scores, and the authors addressed major concerns (assumptions, optimization, scope) in the rebuttal. Therefore, I recommend acceptance as an Oral presentation.

**Reviewer Concerns:**

### Addressed Concerns
The authors provided a comprehensive rebuttal that addressed the majority of the substantive concerns:
- **Misspecification (Assumptions)**: The authors added a significant theoretical extension (Appendix A.5) analyzing the regret bounds under violations of assumptions A1 (perfect USF) and A2 (linear rewards). They complemented this with empirical ablations (Appendix B.5, B.6, B.7), injecting noise into features and rewards, demonstrating robust performance. This successfully addressed the primary concerns of afmy and REKP.
- **Optimization**: A new gradient-based optimization variant was implemented and compared against random shooting (Appendix B.3), showing that random shooting is sufficient and efficient for the dimensionality of typical BFM latent spaces.
- **Experimental Scope**: The authors added experiments on a sparse-reward navigation task (AntMaze) in Appendix B.4, addressing pxZn's question on sparse rewards and eb95's request for non-locomotion tasks.
- ***Labeling Costs**: A new experiment (Appendix B.2) utilizing active learning to request labels only when uncertainty is high was added, directly addressing eb95's concern about real-world deployment costs.

### Remaining Concerns
- **Step-level Theory**: The gap between the theoretical analysis (episodic updates) and the best practical algorithm (step-level updates) remains. The authors acknowledged the difficulty of proving regret bounds for the step-level case due to the complexity of analyzing the trajectory evolution within an episode. However, given the strong empirical evidence and the rigorous analysis of the episodic case, this can be regarded as an acceptable limitation for current work and an interesting avenue for future research.

**Reviewer Scores:**

Based on the high quality of the rebuttal and the engagement during the discussion phase, I estimate the final reviewer sentiment as follows:
- Reviewer afmy(6->8): Given the thorough handling of misspecification and optimization queries, the score would likely rise to 8.
- Reviewer eb95(8->8): The additional experiments and clarifications on real-world use reinforce this positive assessment. The positive score will stand after rebuttal.
- Reviewer REKP(6->8): The detailed theoretical analysis of misspecification directly addressed their main weakness. The score would likely rise to 8.
- Reviewer pxZn(6->8): The clarifications on sparse rewards and baselines were satisfactory. So the score would likely rise to 8.

---

### Decision · Program_Chairs · 2026-01-26

Accept (Oral)